# Tunable high-temperature itinerant antiferromagnetism in a van der Waals magnet

Junho Seo[1,2,10], Eun Su An[1,2,10], Taesu Park[3,10], Soo-Yoon Hwang [4], Gi-Yeop Kim[4], Kyung Song [5], Woo-suk Noh[6], J. Y. Kim[1], Gyu Seung Choi[1,2], Minhyuk Choi[1,2], Eunseok Oh[1,2], Kenji Watanabe [7], Takashi Taniguchi [8], J. -H. Park [2,6], Youn Jung Jo[9], Han Woong Yeom [1,2], Si-Young Choi [4✉], Ji Hoon Shim [2,3✉] & Jun Sung Kim[1,2✉]

Discovery of two dimensional (2D) magnets, showing intrinsic ferromagnetic (FM) or anti-ferromagnetic (AFM) orders, has accelerated development of novel 2D spintronics, in which all the key components are made of van der Waals (vdW) materials and their hetero-structures. High-performing and energy-efficient spin functionalities have been proposed, often relying on current-driven manipulation and detection of the spin states. In this regard, metallic vdW magnets are expected to have several advantages over the widely-studied insulating counterparts, but have not been much explored due to the lack of suitable materials. Here, we report tunable itinerant ferro- and antiferromagnetism in Co-doped $Fe_4GeTe_2$ utilizing the vdW interlayer coupling, extremely sensitive to the material composition. This leads to high $T_N$ antiferromagnetism of $T_N \sim 226$ K in a bulk and ~210 K in 8 nm-thick nanoflakes, together with tunable magnetic anisotropy. The resulting spin configurations and orientations are sensitively controlled by doping, magnetic field, and thickness, which are effectively read out by electrical conduction. These findings manifest strong merits of metallic vdW magnets as an active component of vdW spintronic applications.

[1] Center for Artificial Low Dimensional Electronic Systems, Institute for Basic Science (IBS), Pohang, Korea. [2] Department of Physics, Pohang University of Science and Technology (POSTECH), Pohang, Korea. [3] Department of Chemistry, Pohang University of Science and Technology (POSTECH), Pohang, Korea. [4] Department of Materials Science and Engineering, Pohang University of Science and Technology (POSTECH), Pohang, Korea. [5] Materials Modeling and Characterization Department, KIMS, Changwon, Korea. [6] MPPC-CPM, Max Planck POSTECH/Korea Research Initiative, Pohang, Korea. [7] Research Center for Functional Materials, National Institute for Materials Science, Tsukuba, Japan. [8] International Center for Materials Nanoarchitectonics, National Institute for Materials Science, Tsukuba, Japan. [9] Department of Physics, Kyungpook National University, Daegu, Korea. [10] These authors contributed equally: Junho Seo, Eun Su An, Taesu Park. ✉email: youngchoi@postech.ac.kr; jhshim@postech.ac.kr; js.kim@postech.ac.kr

Van der Waals (vdW) magnets[1–28] offer an intrinsic magnetic multilayer system with various types of magnetic ground states[1–28]. Among them, the interlayer antiferromagnetism with an antiparallel spin configuration across the vdW gap is known to be effectively controlled by magnetic and electric fields, pressure, or doping[7–10]. This leads to various spin-functionalities, particularly for antiferromagnetic (AFM) spintronics, which have gained a lot of attraction recently due to their advantageous properties over ferromagnetic (FM) spintronics, including negligible stray field, robustness against magnetic perturbation, and ultrafast spin dynamics[29–32]. In contrast to the so-called synthetic AFM multilayers, where FM and non-magnetic (NM) layers are stacked alternately using the thin-film deposition techniques[29], vdW antiferromagnets have a highly crystalline and atomically flat interface, free from interface roughness or chemical intermixing. This may lead to highly transparent and uniform magnetic proximity interaction for better spintronic performance in the magnetic multilayers, made of vdW antiferromagnets than synthetic AFM thin films.

One key issue for vdW material-based AFM spintronics is to identify suitable candidate materials. Most of the known vdW antiferromagnets are insulating[4–16], and their interlayer magnetic coupling is mainly through superexchange-like interaction[11–13]. Thus, the modulation of interlayer coupling usually requires structural modification[9–14] and is relatively difficult as compared to synthetic antiferromagnets with the interlayer Ruderman–Kittel–Kasuya–Yosida (RKKY) coupling[29]. Obviously, metallic vdW antiferromagnets[17–20] can be a good alternative. The conduction electrons mediate the interlayer interaction, similar to synthetic antiferromagnets, which is expected to be strongly modulated by changing the composition or the interlayer distance. Furthermore, their longitudinal or transverse conductivities are sensitive to the spin configurations[33–38], offering a direct probe to the spin states even for a few nanometer-thick crystals. Despite these merits, metallic vdW antiferromagnets are rare in nature, except a few recent examples of $GdTe_3$ and $MnBi_2Te_4$ showing a low Neel temperature $T_N < 30$ K[17–20]. Here, we show that an iron-based vdW material, $Fe_4GeTe_2$ with Co doping, hosts the interlayer AFM phase with $T_N \sim 226$ K in a bulk and $\sim 210$ K in 8-nm-thick nanoflakes. Its spin configuration is found to be effectively controlled and read out by conduction electrons, endowing Co-doped $Fe_4GeTe_2$ with a promising role in vdW-material-based spintronics.

## Results

**High-$T_N$ antiferromagnetism in vdW structure.** We consider iron-based metallic vdW ferromagnets $Fe_nGeTe_2$ ($n = 3–5$) as a vdW analog of the synthetic multilayer systems (Fig. 1a)[21–28]. The first known member, $Fe_3GeTe_2$, is experimentally identified as a ferromagnet with $T_c = 220$ K[21–25], but is theoretically

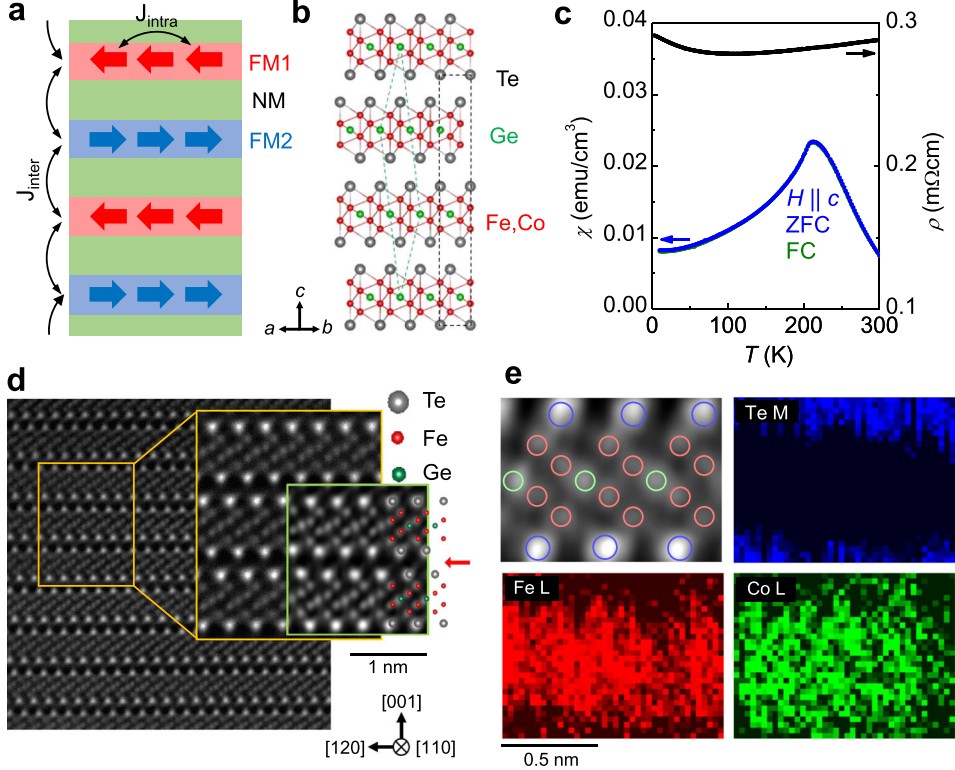

**Fig. 1 Crystal structure and antiferromagnetism of (Fe,Co)$_4$GeTe$_2$. a, b** Schematic illustration of synthetic antiferromagnetic (AFM) multilayers and crystal structure of (Fe,Co)$_4$GeTe$_2$. In both cases, ferromagnetic (FM) layers with intralayer exchange coupling ($J_{intra}$) are stacked alternately and coupled by interlayer coupling ($J_{inter}$). **c** Temperature-dependent magnetic susceptibility $\chi(T)$ under a magnetic field $H = 100$ Oe along the $c$-axis and $\rho(T)$ of (Fe$_{1-x}$Co$_x$)$_4$GeTe$_2$ for $x = 0.33$. A clear cusp in $\chi(T)$, taken during the zero-field-cooling (ZFC) and field-cooling (FC) reveals the AFM transition at $T_N \sim 210$ K. **d** HAADF Scanning transmission electron microscopy (STEM) image of (Fe,Co)$_4$GeTe$_2$ (the first two ones) and Fe$_4$GeTe$_2$ (the last one) crystals along [110]. The STEM image with lower magnification shows that all layers are regularly arranged with a clear vdW gap (red arrow) and without stacking faults throughout whole areas. The comparison of STEM images with higher magnification of (Fe,Co)$_4$GeTe$_2$ (yellow box) and Fe$_4$GeTe$_2$ (green box) shows the almost identical atomic structure. **e** EELS intensity distributions of Te M, Fe L, and Co L edges within the monolayer of (Fe,Co)$_4$GeTe$_2$, indicating the possibilities of finding the corresponding atoms. A large intensity of Te M edge is found at the top and bottom of a monolayer as expected, whereas the intensity for Fe and Co atoms are uniformly distributed between Te layers. This infers that Co dopants successfully substitute Fe atoms as a solid solution.

predicted to be interlayer AFM via RKKY-like interaction[39]. The interlayer AFM coupling is, however, extremely fragile to the small amount of defects or dopants[39], and the AFM phase is mostly inaccessible in real compounds. An alternative candidate is $Fe_4GeTe_2$, recently identified as a high $T_c$ metallic ferromagnet with $T_c = 270$ K[26]. Because of the relatively thick slabs containing two Fe–Fe dumbbells (Fig. 1b), it has stronger intralayer FM coupling than $Fe_3GeTe_2$, together with the interlayer FM coupling, confirmed theoretically and experimentally[26]. By replacing 1/3 of Fe atoms with Co atoms, however, we found that it becomes AFM, as evidenced by its temperature-dependent susceptibility $\chi(T)$ under a magnetic field $H = 100$ Oe along the $c$-axis (Fig. 1c). Hereafter $(Fe,Co)_4GeTe_2$ denotes the AFM compound $(Fe_{1-x}Co_x)_4GeTe_2$ ($x = 0.33$) unless the doping level $x$ is otherwise specified. A clear cusp in $\chi(T)$, taken during the zero-field-cooling (ZFC) and field-cooling (FC), reveals AFM transition at the Neel temperature $T_N = 210$ K. The temperature-dependent resistivity $\rho(T)$ is metallic with an upturn at low temperatures due to Kondo scattering (Fig. 1c and Supplementary Fig. S8). The conductivity of $(Fe,Co)_4GeTe_2$ is $\sim 3 \times 10^5 \, \Omega^{-1} \, m^{-1}$, which is comparable with that of the pristine $Fe_4GeTe_2$[26] and in the bad metal regime (Supplementary Fig. S8). These characteristics make $(Fe,Co)_4GeTe_2$ a unique vdW AFM metal with the high $T_N$ among vdW antiferromagnets (Supplementary Table S1).

The crystal structure of $(Fe,Co)_4GeTe_2$ is the same as $Fe_4GeTe_2$ in a rhombohedral structure (space group $R\bar{3}m$). Scanning transmission electron microscopy (STEM) image of $(Fe,Co)_4GeTe_2$ crystal visualizes the structural units of Fe–Fe dumbbells alternately above and below the plane of Ge atoms (Fig. 1d). These Fe–Fe–Ge–Fe–Fe layers are encapsulated with Te atoms, which shows a similar structural tendency to the pristine $Fe_4GeTe_2$ (the inset in Fig. 1d). A clear vdW gap between the layers is observed without any signature of stacking change or intercalated atoms throughout a wide region. These results imply that Co atoms are dominantly substituted to the Fe sites, not in a type of the interstitial sites. In Fig. 1e, electron energy loss spectroscopy (EELS) analysis visualizes the chemical information within a monolayer, which represents that Co atoms are homogeneously doped in all the Fe sites. Co doping results in the reduction of both the in-plane

$(a = 4.08 \pm 0.04$ Å) and out-of-plane lattice parameters ($c = 29.10 \pm 0.28$ Å), which are deducted from the selected area diffraction pattern (SADP) analysis, in good agreement with X-ray diffraction results (Supplementary Fig. S1).

**Doping-dependent evolution of magnetic phases**. Having established the high-$T_N$ AFM phase in $(Fe,Co)_4GeTe_2$, we focus on the systematic changes of the magnetic and electrical properties of $(Fe_{1-x}Co_x)_4GeTe_2$ single crystals with a variation of Co doping ($0 \le x \le 0.39$). For $x = 0$, the FM transition with in-plane alignment of magnetic moments occurs at $T_c = 270$ K, which is followed by the spin-reorientation transition to the out-of-plane alignment at $T_{SR} = 110$ K (Fig. 2a)[26]. Co doping quickly suppresses the spin-reorientation transition, while keeping the FM transition almost intact with a nearly constant $T_c$ for $x \le 0.23$ (Fig. 2b, c and Supplementary Figs. S2 and S3). Upon further Co doping, however, the AFM order develops from $T_N = 155$ K, well below $T_c$ for $x = 0.26$, and eventually becomes dominant with high $T_N$ up to 226 K for $x = 0.39$ (Fig. 2d–f). The negligible bifurcation between $\chi(T)$ curves taken during ZFC and FC is consistent with the long-range AFM phase. The saturation magnetization $M_{sat}$ monotonically decreases with Co doping from $7.1 \mu_B/f.u.$ ($x = 0$) to $5.5 \mu_B/f.u.$ ($x = 0.39$) due to the magnetic dilution effect (Fig. 2g). Concomitantly, the out-of-plane saturation field $H_{sat}^c$ gradually increases with Co doping in the FM phases for $x < 0.26$. This is consistent with the changes of the magnetic anisotropic energy ($K$) from the easy-axis to the easy-plane types, following $H_{sat}^c = 2K/M_{sat}$ (Fig. 2i). Entering the AFM phase, however, $H_{sat}^c$ is determined by the AFM coupling $J$ as described by $H_{sat}^c \approx 2J/M_{sat}$ and thus suddenly enhanced up to $\sim 6$ T for $x = 0.39$. In the AFM phase, the spin-flop transition is observed for $H\|ab$ at $x = 0.33$, but $H\|c$ at $x = 0.39$ (Fig. 2g and Supplementary Fig. S6), indicating the easy-plane and the easy-axis type spin alignments, respectively. For $x = 0.39$, we found that another magnetic transition occurs to the unknown phase below $T = 90$ K (Fig. 2f and Supplementary Fig. S2). The resulting phase diagram is summarized in Fig. 3a, which manifests that the magnetic configuration and also the magnetic anisotropy of $(Fe_{1-x}Co_x)_4GeTe_2$ are highly sensitive to Co doping.

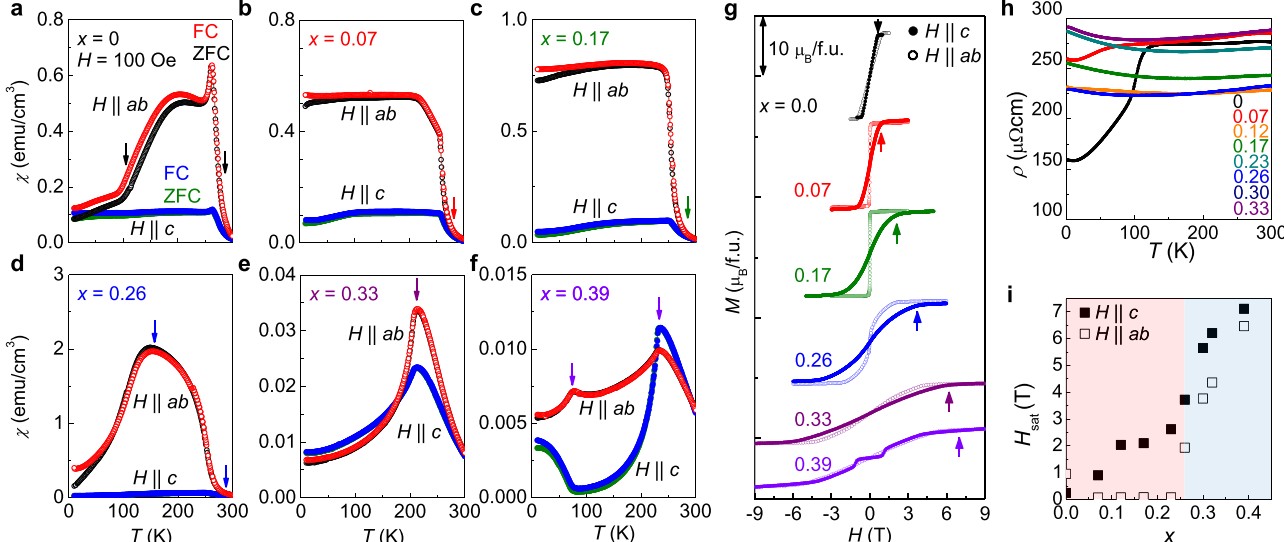

**Fig. 2 Doping-dependent magnetic phase diagram of $(Fe_{1-x}Co_x)_4GeTe_2$. a–f** Temperature-dependent magnetic susceptibility $\chi(T)$ of $(Fe_{1-x}Co_x)_4GeTe_2$ ($0.0 \le x \le 0.39$), taken during zero-field cooling (ZFC) and field-cooling (FC) under $H = 100$ Oe with both magnetic field orientations, $H\|c$ (solid) and $H\|ab$ (open). **g** Magnetization $M(H)$ as a function of magnetic field for $(Fe_{1-x}Co_x)_4GeTe_2$ ($0 \le x \le 0.39$) single crystals, for different field orientations, $H\|c$ (solid) and $H\|ab$ (open). All the $M(H)$ curves were taken at $T = 10$ K, except those for $x = 0.39$ taken at $T = 100$ K. **h** Temperature-dependent in-plane resistivity $\rho(T)$ showing a metallic behavior. **i** The saturation fields $H_{sat}$ for $H\|c$ (solid) and $H\|ab$ (open) as a function of Co doping $x$.

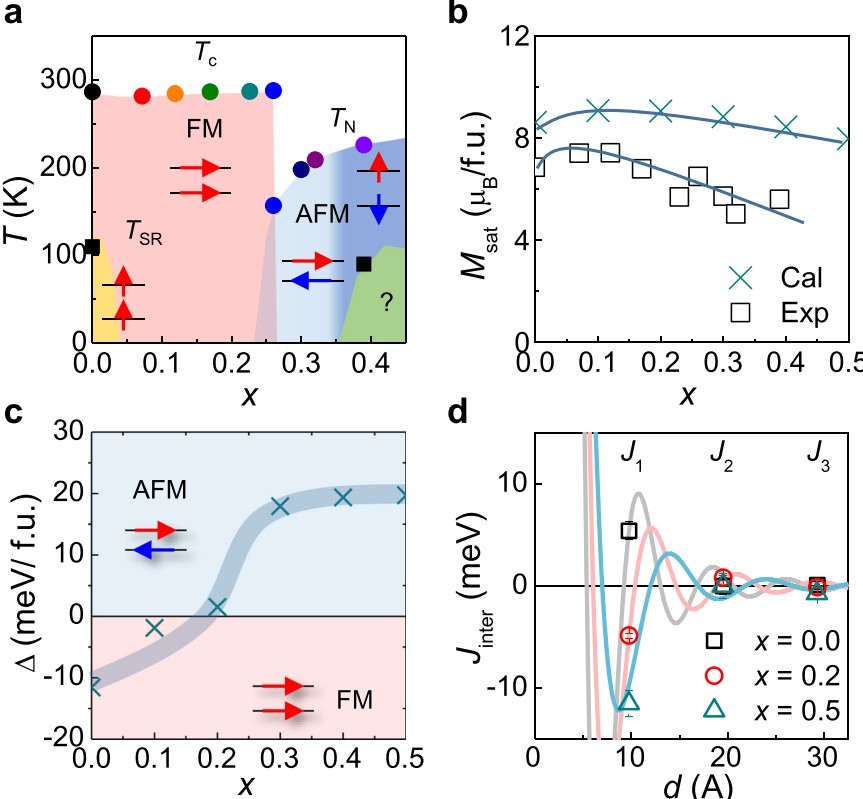

**Fig. 3 First-principles calculations of (Fe$_{1-x}$Co$_x$)$_4$GeTe$_2$. a** Phase diagram of spin states for (Fe$_{1-x}$Co$_x$)$_4$GeTe$_2$ with Curie ($T_C$) and Neel ($T_N$) temperatures. Four different spin states in terms of interlayer spin configurations (FM and AFM) and spin orientations (easy-axis and easy-plane) are stabilized depending on doping level and temperature. The unknown magnetic phase is found for $x = 0.39$ below $T = 90$ K. The schematic illustration of spin configurations is shown in the inset. **b** The saturation magnetization $M_{sat}$ (square symbol) and the calculated $M_{sat}$ (cross symbol) as a function of Co doping $x$. **c** Doping-dependent total energy difference $\Delta$ between the FM and the A-type AFM phases. The FM-to-AFM transition occurs at the critical doping level $x_c \sim 0.2$, in good agreement with experiments. **d** Calculated interlayer exchange interaction $J_{inter}$ with different layer distance ($d$) for $x = 0$, 0.2, and 0.5. The oscillatory dependence of $J_{inter}(d)$, captured by the RKKY model, is systematically changed with Co doping. The errors in the experimental data are smaller than the size of the points.

The first-principles calculations consistently predict the FM-to-AFM phase transition with Co doping. Total energy calculations for the FM and various AFM phases confirm that the FM phase is initially stable at low $x$, but eventually becomes unstable against the AFM phase at high $x$. The most stable AFM phase is the so-called A-type, in which all the spin moments in the whole slab of (Fe,Co)$_4$GeTe$_2$ are ferromagnetically aligned, but across the vdW gap, they are antiferromagnetically coupled. This is consistent with the positive Curie–Weiss temperature from the inverse susceptibility in the AFM phase, suggesting the dominant FM interaction within the layers (Supplementary Fig. S3). Furthermore, from resonant soft X-ray scattering experiments for Fe L3-edge, we observed the magnetic Bragg peak at $q = (0\ 0\ 3/2)$ developed below $T_N$ for the crystal with $x = 0.33$. This confirms the A-type AFM structure, as predicted by calculations (Supplementary Fig. S7). Figure 3c shows the total energy difference $\Delta E = E_{FM} - E_{AFM}$ as a function of Co doping $x$, assuming a random distribution of Co dopants over all Fe sites. The FM-to-AFM transition occurs at the critical doping level $x_c \sim$ 0.2 (Fig. 3a). The estimated $x_c$ as well as $M_{sat}$ from calculations (Fig. 3b, c) is in reasonable agreement with experiments. These results indicate that the evolution of the magnetic phase with Co doping is well captured by first-principles calculations.

Detailed band structure calculations reveal that Co doping affects significantly the density-of-states (DOS) near the Fermi level. In the nonmagnetic calculations, we found a strong DOS peak in the vicinity of the Fermi level (Supplementary Fig. S9).

The resulting strong Stoner instability favors the ferromagnetism within the layer, and the ferromagnetically aligned moment at each layer can be treated as a single localized Heisenberg spin, coupled through the interlayer coupling. This interlayer coupling is, however, determined by a subtle balance of pair exchange interactions between Fe/Co atoms across the vdW gap, which is sensitive to the details in the states at the Fermi level. Using total energy calculations on various interlayer spin configurations (Supplementary Fig. S10) and comparing with the classical Heisenberg Hamiltonian, we extracted the interlayer exchange interaction $J_{inter}$ depending on the layer separation ($d$). We found the oscillatory behavior of $J_{inter}(d)$, well described by the RKKY model[40] (Fig. 3d). The systematic changes of $J_{inter}(d)$ with Co doping, particularly between the nearest-neighboring layers, determine the stability of the AFM phase. These results contrast to the case of insulating vdW antiferromagnets, in which the changes in the interlayer coupling from the FM to AFM types require the stacking modifications[9–14], and manifest the important role of conduction electrons to control the spin configurations of (Fe$_{1-x}$Co$_x$)$_4$GeTe$_2$.

**Electrical detection of spin states.** Conduction electrons are also important to probe the spin state of (Fe,Co)$_4$GeTe$_2$. With magnetic fields along the c-axis, normal to the preferred plane of the spin alignment, the moment at each layer of (Fe,Co)$_4$GeTe$_2$ gradually rotates until its mean direction is parallel to the field at $H_{sat}^c$. This out-of-plane spin canting can be monitored by the

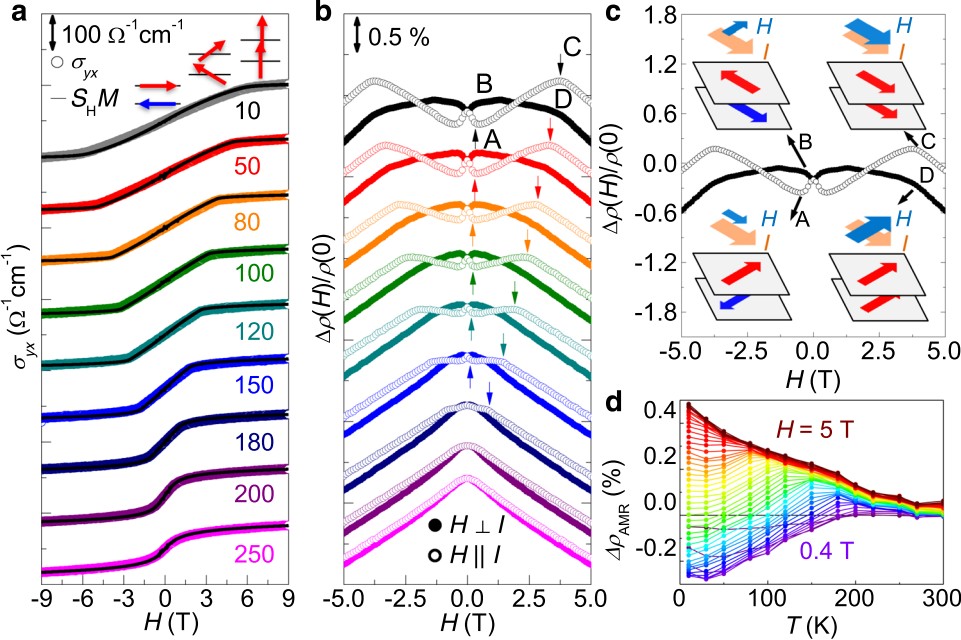

**Fig. 4 Electrical detection of the spin states. a** The transverse conductivity $\sigma_{yx}(H)$ as a function of magnetic fields along the c-axis, taken at various temperatures. The $\sigma_{yx}(H)$ data are nicely reproduced by the field-dependent magnetization $M(H)$ (black solid line) with a scaling factor $S_H \approx 0.3\,V^{-1}$, following the linear relation of $\sigma_{yx}(H) = S_H M(H)$. **b** Magnetoresistance $\Delta\rho(H)/\rho(0)$ under in-plane magnetic fields $H$, parallel (open) or perpendicular (solid) to the current $I$ along the a-axis. The low field spin–flop transition field $H_{sf}^{ab}$ and the high field saturation field $H_{sat}^{ab}$ are indicated by the arrows. **c** Spin configurations with different relative orientations of the magnetic field $H$ and the current $I$. At low $H$, the antiferromagnetically coupled spins are aligned perpendicular to $H$, either $H\|I$ (**a**) or $H{\perp}I$ (**b**). At high $H$, the saturated spins are aligned parallel to $H$, either $H\|I$ (**c**) or $H{\perp}I$ (**d**). **d** Anisotropic magnetoresistance $\Delta\rho_{AMR}$ as a function of temperature and in-plane magnetic field. The low-field and high-field AMR, determined by the relative orientation of Neel vector and the saturated magnetization against the current direction, respectively, which results in a sign change.

anomalous Hall effect (AHE) due to sizable spin–orbit coupling in $(Fe,Co)_4GeTe_2$. As shown in Fig. 4a, the Hall resistivity $\rho_{yx}$ is dominated by the anomalous contribution $\rho_{yx}^A$, i.e., $\rho_{yx} \approx \rho_{yx}^A$, and thus the transverse conductivity $\sigma_{yx} = \rho_{yx}/(\rho_{xx}^2 + \rho_{yx}^2)$ with a variation of magnetic field is nicely scaled with $M(H)$. Moreover, the scaling factor $S_H = \sigma_{yx}/M$ is found to be almost independent of temperature (Supplementary Fig. S5), and thus $\sigma_{yx}(H, T)$ can quantitatively measure the out-of-plane component of net magnetization $M(H, T)$. For example, the magnetic susceptibility, estimated from $\chi^c(T) \propto \sigma_{yx}(H)/H$, allows for experimentally determining $T_N$ in nanoflakes (Fig. 5).

The orientation of the staggered magnetization in the plane, i.e., the Neel vector in the plane can also be effectively tracked by the electrical conductivity. Figure 4b shows the field-dependent magnetoresistance $\Delta\rho(H)/\rho(0)$ of $(Fe,Co)_4GeTe_2$ crystal under different field orientations in the plane, $H\|I$ and $H{\perp}I$ against a current $I\|a$ at various temperatures. Two characteristic fields, $H_{SF}^{ab}$ and $H_{sat}^{ab}$, are identified from clear kinks in both $\Delta\rho(H)/\rho(0)$ curves. The low field $H_{SF}^{ab}$ corresponds to the spin–flop transition, at which the Neel vectors of all domains are fully aligned perpendicular to the in-plane field[33–38]. The Neel vector $L$ is rotated by 90°, from $L{\perp}I$ to $L\|I$, by switching the in-plane magnetic field from $H\|I$ to $H{\perp}I$ (Fig. 4c). This leads to a difference in magnetoresistance due to spin–orbit coupling, which is known as the anisotropic magnetoresistance (AMR) $\Delta\rho_{AMR}^L = \rho_{L\|I} - \rho_{L{\perp}I}$. In $(Fe,Co)_4GeTe_2$, $\Delta\rho_{AMR}^L$ reaches ~0.3%, comparable with other AFM metals[33–36]. With further increasing in-plane field, the moments are canted towards the field, until they are fully aligned at the saturation field $H_{sat}^{ab}$. In this case, the AMR between the cases of $M\|I$ or $M{\perp}I$, $\Delta\rho_{AMR}^M = \rho_{M\|I} - \rho_{M{\perp}I}$ is also expected as found in ferromagnets[41,42]. In $(Fe,Co)_4GeTe_2$, $\Delta\rho_{AMR}^L$

and $\Delta\rho_{AMR}^M$ are comparable in size and opposite in sign, inducing the sign cross of $\Delta\rho_{AMR}$ as summarized in Fig. 4d. Therefore, the AMR allows the electrical access to the orientation of the Neel vector or the saturated magnetization in the plane. Therefore, by combining AHE and AMR we can effectively read out the spin state of $(Fe,Co)_4GeTe_2$.

**Thickness-dependent magnetic phases.** Finally, we discuss the thickness control of the magnetic state of $(Fe,Co)_4GeTe_2$. Owing to the weak vdW interlayer coupling, we obtained nanoflakes with thickness $d$ down to seven layers (7L) for $x = 0.33$ and nine layers (9L) for $x = 0.39$, using mechanical exfoliation. In both cases, the in-plane resistivity $\rho(T)$ shows the metallic behavior (~$10^{-3}\,\Omega cm$) with a low-temperature upturn due to Kondo scattering, similar to the bulk case (Supplementary Figs. S12 and S13). In nanoflakes, we used the transverse conductivity $\sigma_{yx}(H, T)$ to monitor the field and temperature-dependent magnetization $M(H, T)$, as done in the bulk case (Fig. 4a). The resulting susceptibility $\chi^c(T)$ of nanoflakes (Fig. 5e, f and Supplementary Figs. S12 and S13) exhibits a clear kink, indicating the AFM transition at $T_N$ in both cases with $x = 0.33$ and 0.39.

The detailed thickness-dependent phase diagram is different with Co doping level $x$. For nanoflakes with $x = 0.33$, $T$ decreases gradually from the bulk value with lowering thickness. In addition, in nanoflakes, we found a broad hump developed around $T^*$ in $\chi^c(T)$, which shifts to low temperatures with reducing thickness (Fig. 5e). This nonmonotonous temperature dependence of $\chi^c(T)$ implies that spin moments are not fully frozen by dominant AFM interaction in nanoflakes, due to additional competing FM interaction. With further reducing thickness to 7L, this competing FM interaction stabilizes the long-range FM order, evidenced by clear magnetic hysteresis of $\sigma_{yx}(H)$

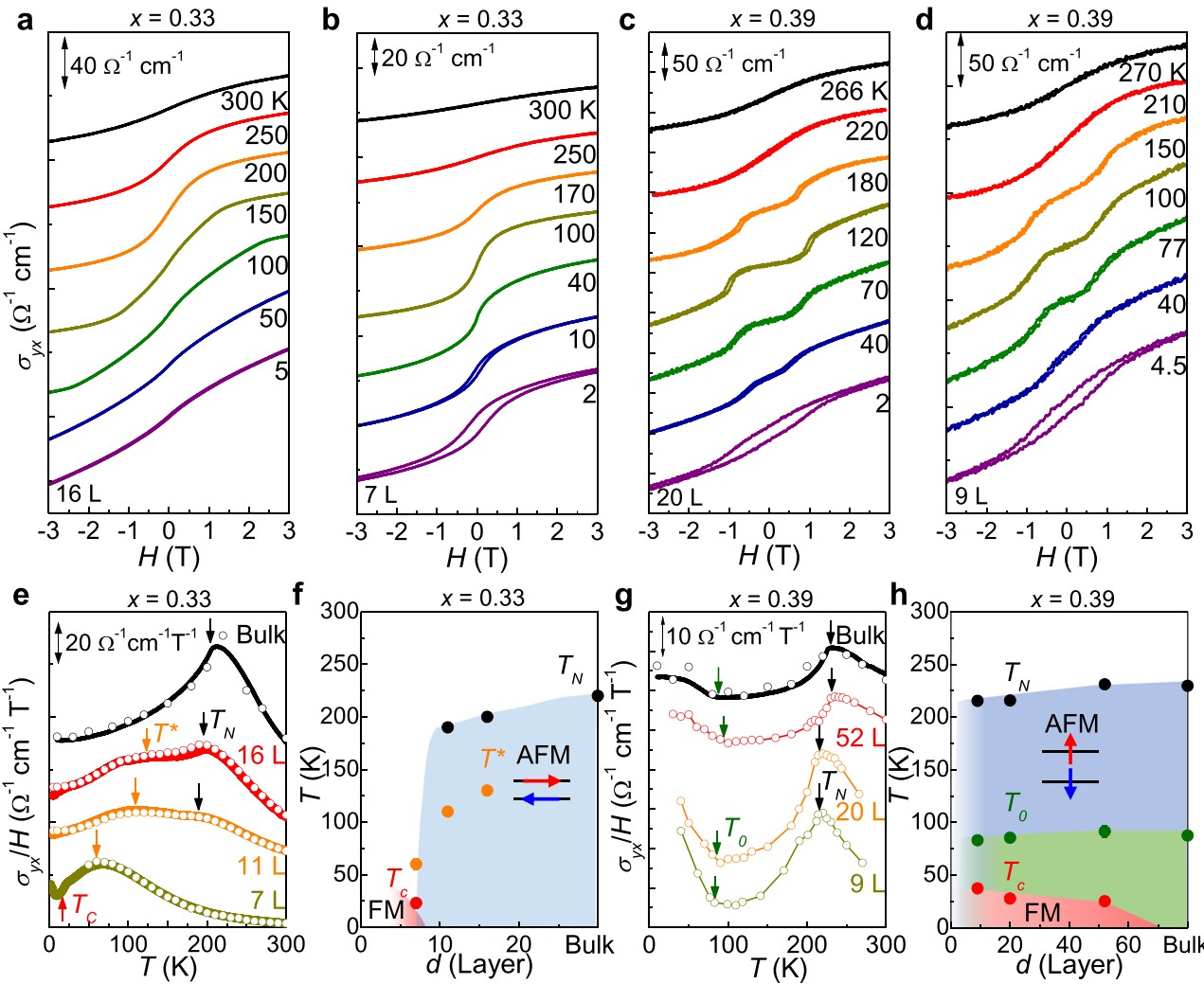

**Fig. 5 Thickness-dependent magnetic phase diagram. a–d** Magnetic field-dependent transverse conductivity $\sigma_{yx}(H)$ at various temperatures of $(Fe_{1-x}Co_x)_4GeTe_2$ nanoflakes ($x = 0.33$) for $H\|c$ in 16L (**a**), 7L nanoflakes (**b**), and $(Fe_{1-x}Co_x)_4GeTe_2$ nanoflakes ($x = 0.39$) for $H\|c$ in 20L (**c**) and 9L nanoflakes (**d**). The $\sigma_{yx}(H)$ curves at low temperatures show typical coercivity in FM phases (**b–d**). **e** Temperature dependence of the magnetic susceptibility $\chi^c(T)$ for $(Fe_{1-x}Co_x)_4GeTe_2$ crystal ($x = 0.33$), estimated from the low-field slope of $\sigma_{yx}(H)$ at each temperature (open circle) or from the difference between $\sigma_{yx}(T)$ curves taken at $H = \pm0.1$ T for $H\|c$ (solid circle). Magnetic transition temperature $T_N$ and $T_C$ are indicated by the arrows, together with the characteristic temperature $T^\star$, determined by a broad hump in $\chi^c(T)$. **f** Thickness-dependent magnetic phase diagram of $(Fe_{1-x}Co_x)_4GeTe_2$ crystal ($x = 0.33$) with characteristic temperatures of $T_N$, $T^\star$, and $T_c$. **g** Temperature dependence of the magnetic susceptibility $\chi^c(T)$ for $(Fe_{1-x}Co_x)_4GeTe_2$ crystal ($x = 0.39$) (open circle), bulk magnetization with $H = 0.1$ T for $H\|c$ (solid symbol). Characteristic temperature $T_0$ is indicated by the arrows at the inflection of $\chi^c(T)$ at low temperature. **h** Thickness-dependent magnetic phase diagram of $(Fe_{1-x}Co_x)_4GeTe_2$ crystal ($x = 0.39$) with characteristic temperatures of $T_N$, $T_0$, and $T_c$. The errors in the experimental data are smaller than the size of the points.

with a coercive field $H_c = 0.21$ T at $T = 2$ K (Fig. 5b). The $H_c$ gradually decreases with increasing temperature up to $T_c \approx 25$ K (Supplementary Fig. S12). The corresponding $\chi^c(T)$ exhibits no signature of AFM ordering but only a broad peak at $T^\star \sim 40$ K. The resulting phase diagram (Fig. 5g) reveals that thickness tuning offers another effective means to tune the interlayer FM and AFM interactions, even though the 2D limit is yet to be reached. It has been well established that vdW crystals expand along the $c$-axis by ~0.3–0.7%,[43,44] when they are thinned to be tens of nanometer thick. Considering that the $c$-axis lattice constant shrinks by ~0.6% with Co doping of $x = 0.39$, this thinning-induced swelling affects significantly the magnetic ground state near the FM–AFM phase boundary.

In nanoflakes with higher Co doping of $x = 0.39$, located deep inside the AFM phase of the phase diagram (Fig. 3a), the AFM phase is expected to be more stable than the nanoflakes with $x =$

0.33. As shown in Fig. 5, this is indeed the case. Upon reducing the thickness down to 9L, we observed similar $\sigma_{yx}(H)$ curves with those of the bulk (Fig. 2g and Supplementary Fig. S5d). $T_N$ is also reduced slightly to ~210 K from its bulk value of 226 K, and the low-temperature transition at $T_0$ is also maintained without significant change. At low temperatures, we observed the magnetic hysteresis (Fig. 5c, d and Supplementary Fig. S13), indicating the FM phase, as found in the case of $x = 0.33$. The resulting thickness-dependent phase diagram for $x = 0.39$ reveals that the AFM phase in the intermediate temperature range between $T_0$ and $T_N$ is stable in nanoflakes. The spin–flop transition field $H_{SF}$, however, is systematically reduced with lowering thickness (Supplementary Fig. S13f). These results clearly demonstrate that the stability of the AFM phase and the magnetic anisotropy can be controlled by thickness and doping levels in $(Fe_{1-x}Co_x)_4GeTe_2$.

## Discussion

Our observations unequivocally show that $(Fe,Co)_4GeTe_2$ is an intrinsic high-$T_N$ AFM multilayers. Although its $T_N$ is still below room temperature, an order of magnitude enhancement of $T_N$, as compared to other metallic vdW antiferromagnets[17–20], is achieved by tuning the vdW interlayer coupling between the strongly FM layers of $Fe_4GeTe_2$. This approach can be applied to other recently discovered high-$T_c$ vdW ferromagnets[26–28, 42] to possibly realize the room temperature antiferromagnetism. Furthermore, $(Fe_{1-x}Co_x)_4GeTe_2$ hosts at least four different spin states in terms of interlayer spin configurations (FM and AFM) and spin orientations (in-plane and out-of-plane) while keeping the same stacking structure (Fig. 3a). The switching between these states, demonstrated using doping, magnetic field, and thickness control, can be more effectively achieved in vdW heterostructures through, e.g., the exchange-spring effect[45, 46] with an adjacent FM vdW layer or the current-induced spin–orbit torque[34, 37] with large SOC materials, as done in metal spintronics. The intimate coupling of the spin states to electrical conduction in $(Fe,Co)_4GeTe_2$ (Fig. 4a, b) ensures the electrical readout of the spin state. These strong controllability and readability on the spin states are the manifestation of itinerant magnetism, highly distinct from insulating vdW magnets. We, therefore, envision that metallic vdW antiferromagnets, including $(Fe,Co)_4GeTe_2$ in this work, will enrich the material candidates and the spin functionalities for all vdW material-based spintronics.

## Methods

**Single-crystal growth and characterization.** Single crystals were grown by a chemical vapor transport method using pre-synthesized polycrystalline sample and iodine as a transport agent. The obtained single crystals had plate-like shape with a typical size of ~$1 \times 1 \times 0.04$ mm$^3$. The high crystallinity of single crystals was confirmed by X-ray diffraction (Supplementary Fig. S1). From the energy-dispersive spectroscopy measurements, we confirmed a systematic variation of Co doping $x$, in the presence of Te deficiency by ~10% and excess of the total (Fe, Co) content by ~5%, similar to pristine $Fe_4GeTe_2$[26]. Magnetization was measured under magnetic field along the $c$-axis or $ab$-plane using a superconducting quantum interference device magnetometer (MPMS, Quantum Design) and vibrating sample magnetometer option of Physical Property Measurement System (PPMS-14T, Quantum Design). The in-plane resistivity and the Hall resistivity were measured in the standard six probe configuration using a Physical Property Measurement System (PPMS-9T, Quantum Design).

**Device fabrication.** Using mechanical exfoliation in the inert argon atmosphere ($H_2O < 0.1$ p.p.m., $O_2 < 0.1$ p.p.m.), we obtained nanoflakes of $(Fe,Co)_4GeTe_2$ on top of Si/SiO$_2$ substrate, pre-treated by oxygen plasma ($O_2 = 10$ s.c.c.m., $P \sim 100$ mTorr) for 5 min to remove adsorbates on the surface. The exfoliated crystal is then subsequently covered by a thin h-BN flake to prevent oxidation during device fabrication. Typically $(Fe,Co)_4GeTe_2$ flakes are of several tens of μm$^2$ in the area and down to ~7 nm (~7L) in thickness, estimated by the atomic force microscopy measurements (Supplementary Fig. S11). To make electrodes for transport measurements, we employed an electron beam lithography technique, using poly (methyl methacrylate)-positive resist layer, which was spin-coated and dried in vacuum at room temperature. After etching, the patterned area of the covered h-BN flake with CF$_4$ plasma, Cr(10 nm)/Au(50 nm), is deposited on the exposed surface of the nanoflake.

**STEM and EELS.** STEM samples were made along [1 1 0] projections, which show the most distinguishable atomic structures. Samples were prepared with dual-beam focused ion beam systems (Helios and Helios G3, FEI). Different acceleration voltage conditions from 30 to 1 keV were used to make a thin sample with less damages. The subsequent Ar ion milling process was performed with low energy (PIPS II, Gatan Inc., USA). The atomic structure was observed using a STEM (JEM-ARM200F, JEOL, Japan) at 200 kV equipped with a fifth-order probe corrector (ASCOR, CEOS GmbH, Germany) at Materials Imaging & Analysis Center of POSTECH in Republic of Korea. The electron probe size was ~0.8 Å, and the high-angle annular dark-field detector angle was fixed from 68 to 280 mrad. The SADP was obtained using a TEM (JEM-2100F, JEOL, Japan) at 200 kV equipped with a spherical aberration corrector (CEOS GmbH, Germany). The raw STEM images were compensated with ten slices by SmartAlign and processed using a band-pass difference filter with a local window to reduce background noise (SmartAlign and Filters Pro, HREM Research Inc., Japan). For EELS analysis, we utilized another STEM (JEM-2100F, JEOL) with a spherical aberration corrector

(CEOS GmbH) equipped with an EEL spectrometer (GIF Quantum ER, Gatan, USA). The used probe size was ~1.0 Å under 200 kV and the chemical analysis was performed by using a Spectrum Image via the STEM mode. The obtained spectral data from a spectral image were filtered to intensify the Fe-, Co-, and Te-edge signals by MSA (Multivariate Statistical Analysis, HREM Research Inc., Japan).

**Scanning tunneling microscopy.** The surface of $(Fe,Co)_4GeTe_2$ was characterized by the scanning tunneling microscopy (STM) on a single crystal. The single crystal is first cleaved in a high vacuum (~$1 \times 10^{-8}$ torr) to obtain the clean surface and then is transferred to ultra-high vacuum (~$1 \times 10^{-11}$ torr) for the STM measurements at 77 K.

**First-principles calculations.** Electronic structure calculations were performed using a full-potential linearized augmented plane wave method, implemented in WIEN2K package[47]. Experimental lattice constants of $(Fe,Co)_4GeTe_2$ are used and exchange-correlation potential is chosen to the generalized gradient approximation of Perdew–Berke–Ernzerhof[48]. Assuming that Co is doped at all Fe sites homogeneously throughout the $Fe_4GeTe_2$ layer, the virtual crystal approximation is considered in the DFT calculation of $(Fe,Co)_4GeTe_2$.

**Soft X-ray scattering.** Resonant soft X-ray scattering experiment was carried out in 6A MPK MeXiM beamline, PAL. The single crystals were cleaved in the air and thereafter immediately transferred to the chamber with the ultra-high vacuum pressure of ~$8 \times 10^{-9}$ torr. Photon energy was selected near the Fe L3-edge absorption energy value at 707.5 eV, because $q = (0\ 0\ 3/2)$ peak has a different energy profile with $(0\ 0\ 3)$ Bragg peak.

## Data availability

The data that support the findings of this study are available from the corresponding authors on request.

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

## Acknowledgements

We thank H.W. Lee, K. Kim, M.H. Jo for fruitful discussion. We also thank H. G. Kim in Pohang Accelerator Laboratory (PAL) for the technical support. This work was supported by the Institute for Basic Science (IBS) through the Center for Artificial Low Dimensional Electronic Systems (no. IBS-R014-D1), by the National Research Foundation of Korea (NRF) through SRC (Grant No. 2018R1A5A6075964), the Max Planck-POSTECH Center for Complex Phase Materials (Grant No. 2016K1A4A4A01922028), and also by the NRF (No. NRF-2019R1A2C1089017, No. 2020R1A5A1019141, and No. 2020M3H4A2084418). S.-Y.C. acknowledges the support of the Global Frontier Hybrid Interface Materials by the NRF (Grant No. 2013M3A6B1078872). K.W. and T.T. acknowledge support from the Elemental Strategy Initiative conducted by the MEXT, Japan, Grant Number JPMXP0112101001, JSPS KAKENHI Grant Number JP20H00354, and the CREST (JPMJCR15F3).

## Author contributions

J.S.K., E.S.A., and J.S. conceived the experiments. J.S. synthesized the bulk crystals. E.S.A. and G.S.C. carried out device fabrication and measurements on nanoflakes. J.S., M.C., and Y.J.J performed the transport property measurements on bulk crystals. T.P. and J.H. S. performed the electronic structure calculations and the analysis. S.-Y.H., G.-Y.K., K.S., and S.-Y.C. carried out structural and chemical identification using scanning transmission electron microscopy and electron energy loss spectroscopy. W.-s.N., J.Y.K., and J.-H. P. carried out resonant soft X-ray scattering experiments. E.O. and H.W.Y. performed surface characterization using scanning tunneling microscopy measurements. K.W. and T.T provided boron nitride crystals. J.S., E.S.A., T.P., S.-Y.C., J.H.S., and J.S.K. co-wrote the manuscript. All authors discussed the results and commented on the paper.

## Competing interests

The authors declare no competing interests.
