## [Peer Review File · Nature Communications]

Reviewers' Comments:

Reviewer #1:

Remarks to the Author:

Searching vdW AFM with high Neel temperature is an interesting and important research topic. However, the results of magnetic and electric transport measurements shown in the paper cannot confirm the AFM state. More experimental results are needed. I list several comments below.

1. Figure 1c, The magnetic susceptibility peak near 220 K cannot guarantee a AFM state. Special spin cluster behaviour may also generate a similar peak.
2. Figure 2a, The evolution of magnetic loops cannot guarantee a AFM state when the doping is 0.39. It may due to the formation of a large in-plane magnetic anisotropy.
3. Fig 3a, We can easily explain the data using a ferromagnetic state with large in-plane magnetic anisotropy at low temperature. With increasing temperature, the magnetic anisotropy decreases. Hence, the magnetization can be saturated at lower field at high temperature.
4. Fig 3b, 3c, Using in-plane ferromagnetic state, you can also explain the transport results.
5. Fig 4c and 4d, thicker sample has lower coercivity can be explain quite easily. The reason is that the domain can expand more easily when sample becomes thicker.

6. The R(T) curves can have many different explanations.

I understand the authors did calculations to show the AFM state. However, it is quite often that the calculations results are different from experimental results. From the magnetic and electric transport results, we cannot rule out the material is still in FM state (a FM state with strong in-plane anisotropy).

I think that the authors need to do one more experiment, a neutron diffraction on the bulk crystals. If the neutron diffraction confirms the AFM structure, I will fully support the publication of this paper in Nature Communications.

If the neutron diffraction shows a FM structure, the impact of this paper is NOT high enough for Nature Communications.

Reviewer #2:

Remarks to the Author:

This manuscript describes a doping study of the metallic vdW ferromagnet Fe_4GeTe_2 , where it is successfully tuned to an AFM. I think this paper is worthy of publication in net. Comm. But it needs a bit more work. While I really enjoyed reading the introduction, the description of the results is hard to follow and I have several comments for improvements. I also think that, at least for some doping concentrations more magnetic susceptibility data would be helpful. The biggest issue with this manuscript it that the presentation is confusing and I would therefore recommend a second round of revision, where it might be easier to understand.

1) The text does not always refer to the SI when necessary. For many statements in the I had to search for a while to find the data that corroborate that statement. This made reviewing this paper very challenging. I recommend that the authors read their manuscript carefully and make sure they refer the reader to the relevant figure whenever it is described. On a similar note, figure captions can be improved. For example, nowhere in Fig. S2 is the applied field giving, In the main text I found that it is supposedly 1kOe, but in this sentence there is no reference to Figure S2. This is very confusing and hard to follow.

- 2) I think Fig 2 should be modified. Fig. 2a does not really show the reader that the phase diagram in Figure 2g is correct. This is better seen with the data from Fig. S2, which I think might be better placed to the main text. In Fig 2a, it appears as if there is a smooth change in magnetization with doping and one cannot see any clean transition from FM to AFM. It took me a while to realize that the magnetic phase diagram is correct. This is also because of the lack of reference to the SI when starting to describing the trends as discussed above. I think in the current stage, the reader will need a long time to understand the results.
- 3) For the higher doped samples, it might be nice to take M vs T curves at more applied fields to better understand the appearance of the unknown magnetic phase.
- 4) The authors motivate their study with 2D magnets and say there is no 2D metallic AFM with a high Neel temperature yet. That is true. However, the authors also did not introduce a 2D one, since their result suggest that the samples become FM when thin. I think they should be more honest about that fact in the abstract and intro, by inserting the word "bulk" before they discuss their results of a metallic vdW AFM.
- 5) Can the authors comment on the upturn in resistivity at lower T? Why is this? Why is it opposite to the trend in FM samples?
- 6) I think the authors should discuss the role of disorder in their samples and how this might affect the properties. This might also help for the discussion in respect to point 5.
- 7) For the calculations, the authors say they used VCA to model the doping, but they do not say how the lattice constants were obtained. Did they use experimental ones? And if yes in accordance with change in doping? And if not did they optimize the structures? And if yes, did they correct for vdW forces? A few more details here would be helpful.

Reviewer #3:

Remarks to the Author:

This work presents the evolution of magnetic properties in Fe_4GeTe_2 with Co substitution on Fe site.

I will start with some technical comments.

1. Authors have synthesized single crystals, apparently. Therefore, I am puzzled why anisotropy in magnetic properties has not been presented?
2. Supplement section Fig. S1a: crystals have small impurity phase, between the strongest peak around $2\theta=27$ and (221) there is another reflection in all crystals. Then, $x=0.39, 0.3, 0.17$ also contain another visible peak between (221) and (444). Since many results in this work rest on bulk magnetic characterization it is of interest to index those peaks and, if due to impurity phases, ensure that bulk magnetic properties are not influenced by impurities.
3. Authors should show saturated moment in $\mu_B/\text{F.U.}$ where F.U. is formula unit in Fig. 2a in the main text as well as in Fig. S3(a-e). There seems to be a misprint in Fig. 2a, 0.7 is in fact 0.07.
4. I would also show ZFC-FC $\chi(T)$ curves. Can authors rule out spin glass state in all investigated crystals?
5. I find Fig. 1b rather interesting...I wonder what is the Ioffe-Regel limit for AFM crystals? Note the absence of metallic resistivity for high x at low temperatures. Throughout the text authors refer to investigated crystals as "metals". Yet, $\rho(T)$ looks like a very bad metal for high x. I would advise the use of more precise language.
6. In Fig S5 authors report change in DOS with Co doping. Therefore, it is reasonable to assume some change in electronic structure. Does that have any effect on electrical detection of the spin state. For example, could changes in AMR be influenced by electronic structure changes, in addition to Neel vector changes?

I do not think that this work brings high impact or innovation. I agree that AFM transition is high but the last sentence in the introduction is greatly exaggerated. Promising material in spintronics can not operate at 230 K. Authors also say in the conclusion that $(\text{Fe,Co})_4\text{GeTe}_2$ will enrich the material candidates and spin functionalities for...spintronics. I do not see how. Thickness-dependent magnetic transition falls down with size reduction therefore it is difficult to think of

future devices based on nanofabricated crystals investigated in this work. On the other hand, since this paper does report new incremental knowledge, it might be suitable for different, more specialized journal if technical comments above are properly addressed.

Reply to the Reviewers

Answers to reviewer #1's questions

We sincerely appreciate the reviewer's helpful comments and suggestions. We did our best to answer to each single question and we hope our answers satisfy the reviewer.

Q1-1. Figure 1c, The magnetic susceptibility peak near 220 K cannot guarantee a AFM state. Special spin cluster behaviour may also generate a similar peak.

A1-1. As the reviewer #1 pointed out, the transition to the spin cluster or spin-glass phase can produce a magnetic susceptibility peak, which can be misinterpreted as a signature of the AFM phase transition. In order to rule out the possibility of the spin-cluster state, we measured the temperature dependent magnetic susceptibility, $\chi(T)$, at different cooling procedures, zero-field cooling (ZFC) and field cooling (FC). It has been well known that for the spin-cluster state, the peak of $\chi(T)$ is accompanied by a pronounced bifurcation between ZFC and FC curves. In the $(\text{Fe}_{1-x}\text{Co}_x)_4\text{GeTe}_2$ single crystal with $x = 0.33$, however, we observed negligible difference between the ZFC- and FC- $\chi(T)$ curves for both $H \parallel c$ and $H \parallel ab$ as shown in Fig. 2 of the revised manuscript. These results indicate that the peak in $\chi(T)$ for the crystal with $x = 0.33$ reflects the transition, not to the spin-cluster phase, but to the antiferromagnetic (AFM) phase.

Consistently we also observed no signature of the relaxation behavior, one of the hallmark of the spin cluster state. In the spin cluster phase the change of magnetization requires a sizable time, leading to the time-dependent magnetization phenomena. For example, in the ferromagnetic phase in $(\text{Fe}_{1-x}\text{Co}_x)_4\text{GeTe}_2$ crystals with low x , we found a relaxation behavior with a characteristic time scale of $\tau \sim 600$ sec (Supplementary Fig. S4), which is comparable

FIG. 2. Doping dependent magnetic phase diagram of $(\text{Fe}_{1-x}\text{Co}_x)_4\text{GeTe}_2$. **a-f**, Temperature dependent magnetic susceptibility $\chi(T)$ for $(\text{Fe}_{1-x}\text{Co}_x)_4\text{GeTe}_2$ ($0 \leq x \leq 0.39$) single crystals, with magnetic field $H = 100$ Oe for different orientations and cooling process, $H \parallel ab$ (open) zero-field-cooling (black), field-cooling (red) and $H \parallel c$ (solid) zero-field-cooling (green), field-cooling (blue). **g**, Magnetization $M(H)$ as a function of magnetic field for $(\text{Fe}_{1-x}\text{Co}_x)_4\text{GeTe}_2$ ($0 \leq x \leq 0.39$) single crystals, for different field orientations, $H \parallel c$ (solid) and $H \parallel ab$ (open). All the $M(H)$ curves were taken at $T = 10$ K, except those for $x = 0.39$ taken at $T = 100$ K. **h**, Temperature dependent in-plane resistivity $\rho(T)$ showing a metallic behavior. **i**, The saturation fields H_{sat} for $H \parallel c$ (solid) and $H \parallel ab$ (open) as a function of Co doping x .

FIG. S4. **Thermoremanent magnetization relaxation in $(\text{Fe}_{1-x}\text{Co}_x)_4\text{GeTe}_2$.** **a**, Magnetic field dependent magnetization for $x = 0.07$ at $T = 10$ K. Clear hysteresis behavior with coercive field $H_c \sim 6$ Oe is observed. **b,c**, Thermoremanent magnetization relaxation for $x = 0.07$ at $T = 10$ K with waiting time $t_w = 193$ s (**b**) and 1350 s (**c**). The red solid line is the best fit of the relaxation model. **d**, Magnetic field dependent magnetization for $x = 0.33$ at $T = 10$ K. In contrast to $x = 0.07$ case, no hysteresis behavior is observed. **e,f**, Thermoremanent magnetization relaxation for $x = 0.33$ at $T = 10$ K with waiting time $t_w = 176$ s (**e**) and 1010 s (**f**). No signature of relaxation behavior is observed for $x = 0.33$.

with the spin cluster phases in, e.g. PrRhSn_3 and $\text{Sr}_2\text{Mn}_{0.7}\text{Fe}_{0.3}\text{MoO}_6$ [Anand, V. K. *et al. Phys. Rev. B* **85**, 014418 (2012), Wang, X. *et al. J. Appl. Phys.* **109**, 07C322 (2011)]. For $x = 0.33$, however, we found no signature of relaxation behavior as shown in Supplementary Fig. S4 in the revised manuscript.

Finally, in order to present a more direct experimental evidence of the AFM ordering, we performed resonant soft X-ray scattering experiments for the crystal with $x = 0.33$, as shown in the supplementary Fig S7. Similar to neutron diffraction, resonant X-ray scattering also offer

FIG. S7. **Resonant soft X-ray scattering of $(\text{Fe}_{0.67}\text{Co}_{0.33})_4\text{GeTe}_2$.** **a**, Resonant soft X-ray scattering intensity at 300 K ($> T_N$) and 70 K ($< T_N$). In addition to Bragg peak at $q = (0 0 3)$, additional peak at $q = (0 0 3/2)$ is developed below T_N . **b**, Crystal structure of $(\text{Fe}_{0.67}\text{Co}_{0.33})_4\text{GeTe}_2$ with the interlayer AFM structure, indicated by red and blue shades. The period of 19.5 Å, corresponding to $q = (0,0,3/2)$, is consistent with the interlayer AFM phase

a probe for AFM phase. In addition to the Bragg peaks at (003), a clear additional peak develops below T_N at $q = (0,0,3/2)$, which is absent above T_N . The corresponding periodicity is $\sim 19.5 \text{ \AA}$ along the c -axis, in excellent agreement with the interlayer AFM (A-type) structure, predicted by our first principle calculations. Although further studies, using neutron diffraction, are helpful to determine the size and orientation of magnetic moments in the AFM phase of $(\text{Fe}_{1-x}\text{Co}_x)\text{GeTe}_2$, our resonant X-ray scattering results provide a direct evidence of the AFM phase in $(\text{Fe}_{1-x}\text{Co}_x)_4\text{GeTe}_2$ with $x = 0.33$.

As the reviewer #1 requested, we added several sentences and additional Figures (Fig. 2) in the main text and Supplementary Information (Figs. S4, and S7), in order to provide experimental evidence of the long-range AFM phase.

Q1-2. *Figure 2a, The evolution of magnetic loops cannot guarantee a AFM state when the doping is 0.39. It may due to the formation of a large in-plane magnetic anisotropy.*

A1-2. We agree with the reviewer that the magnetization loop alone cannot guarantee the AFM state for $(\text{Fe}_{1-x}\text{Co}_x)_4\text{GeTe}_2$ crystal. As explained in **A1-1**, we provide several experimental evidence of the AFM transition for $x = 0.33$, including the temperature dependent magnetic susceptibility $\chi(T)$ with ZFC and FC. As explained for the crystal with $x = 0.33$ in **A1-1**, we also performed the magnetic susceptibility measurements on the crystal with $x = 0.39$ under ZFC and FC procedures for both in $H // c$ and $H // ab$ (Fig. 2f of the revised manuscript). We found almost negligible bifurcation between ZFC- and FC- $\chi(T)$ curves in the temperature range of $T_0 = 75 \text{ K} < T < T_N = 226 \text{ K}$. Furthermore, for $H // c$, a clear spin-flop transition is observed in the field-dependent magnetization curve $M(H)$ at $H = 1 \text{ T}$, well below the saturation field of $H_{\text{sat}} = 7 \text{ T}$ as shown in Fig. 2g and the supplementary Fig. S6. This spin-flop transition cannot occur in the FM phase. These results strongly suggest that the AFM phase is stabilized in the intermediate temperature range $T_0 = 75 \text{ K} < T < T_N = 226 \text{ K}$.

Below $T_0 = 75 \text{ K}$, however, we found a small bifurcation between ZFC- and FC- $\chi(T)$ curves. This confirms that magnetic structure at low temperatures for $x = 0.39$ is distinct from the interlayer AFM phase and indicates the competing FM interaction may play a role. In order to identify the low-temperature magnetic structure of the crystal with $x = 0.39$, the detailed study is needed, which however is beyond the scope of this work.

Q1-3. *Fig 3a, We can easily explain the data using a ferromagnetic state with large in-plane magnetic anisotropy at low temperature. With increasing temperature, the magnetic anisotropy decreases. Hence, the magnetization can be saturated at lower field at high temperature. Fig 3b, 3c, Using in-plane ferromagnetic state, you can also explain the transport results.*

A1-3. As explained in **A1-1**, our resonant X-ray scattering provide an experimental evidence for the interlayer AFM phase in $(\text{Fe}_{1-x}\text{Co}_x)_4\text{GeTe}_2$ with $x = 0.33$. Considering the in-plane magnetic anisotropy of the system, as pointed out by the reviewer, together with the interlayer AFM structure, one can explain the Hall effect and anisotropic magnetoresistance (AMR) results in terms of the canting of moments, as shown illustrated in Fig. 4 of the revised main text. Particularly, under the in-plane magnetic field, the two characteristic kinks at H_1 and H_2

FIG. 4. Electrical detection of the spin states. **a**, The transverse conductivity $\sigma_{yx}(H)$ as a function of magnetic fields along the c axis, taken at various temperatures. The $\sigma_{yx}(H)$ data are nicely reproduced by the field dependent magnetization $M(H)$ (black solid line) with a scaling factor $S_H \approx 0.3 \text{ V}^{-1}$, following the linear relation of $\sigma_{yx}(H) = S_H M(H)$. **b**, Magnetoresistance $\Delta\rho(H)/\rho(0)$ under in-plane magnetic fields H , parallel (open) or perpendicular (solid) to the current I along the a axis. The low field spin-flop transition field $H_{\text{sf}}^{\text{ab}}$ and the high field saturation field $H_{\text{sat}}^{\text{ab}}$ are indicated by the arrows. **c**, Spin configurations with different relative orientations of the magnetic field H and the current I . At low H , the antiferromagnetically coupled spins are aligned perpendicular to H , either $H \parallel I$ (A) or $H \perp I$ (B). At high H , the saturated spins are aligned parallel to H , either $H \parallel I$ (C) or $H \perp I$ (D). **d**, Anisotropic magnetoresistance $\Delta\rho_{\text{AMR}}$ as a function of temperature and in-plane magnetic field. The low-field and high-field AMR, determined by the relative orientation of Neel vector and the saturated magnetization against the current direction, respectively, which results in a sign change.

are consistent with this AFM structure. At low magnetic fields, the Neel vector of each domain is fully aligned at H_1 , producing a kink in the AMR. With further increasing magnetic fields above H_1 , canting of magnetic moments along the external field occurs, until all the moments are fully aligned at H_2 . In the case of the in-plane ferromagnets, however, we expect a single anomaly at the saturation field without a low-field kink. Together with resonant X-ray scattering results, our Hall resistivity and AMR data demonstrate that the spin orientation and configuration can be read out by electrical conduction.

Q1-4. Fig 4c and 4d, thicker sample has lower coercivity can be explain quite easily. The reason is that the domain can expand more easily when sample becomes thicker.

A1-4. As explained in A1-1, the magnetic susceptibility curves for the bulk crystal with $x=0.33$ taken during zero-field cooling (ZFC) and field-cooling (FC), show nearly identical temperature dependence. These results rule out the possible spin cluster or spin glass state and are consistent with the resonant X-ray scattering results shown in Supplementary Fig. S7. Similarly, in the nanometer-thick crystals, we observed a clear peak and no thermal hysteresis

in the magnetic susceptibility $\chi(T)$, extracted from the Hall resistivity of the nanoflakes with thickness of 11 L and 16 L. These results indicate the AFM phase is stable in these nanoflakes, while the FM phase appears in thinner crystal with 7 L, as shown in Fig. 5 and Supplementary Fig. S12.

Furthermore, in the revised manuscript, we present new results on the thickness-dependence of the high Co-doped crystal with $x = 0.39$. In this case, we found that the AFM phase is stable in nanoflakes with thickness down to ~ 9 L as shown in Fig. 5 of the revised manuscript. Using the same method for the case of $x = 0.33$, we estimate the magnetic susceptibility $\chi(T)$ from the Hall resistivity of nanoflakes with $x = 0.39$. The obtained $\chi(T)$ shows a sharp peak due to the AFM transition at $T_N = 226$ K, followed by a clear upturn at $T_0 = 75$ K, which is similar to the bulk case. Furthermore, a spin-flop transition is clearly observed in the field dependent Hall resistivity of nanoflakes in the intermediate temperature range $T_0 = 75$ K $< T < T_N = 226$ K

FIG. 5. Thickness-dependent magnetic phase diagram. **a-d**, Magnetic field dependent transverse conductivity $\sigma_{yx}(H)$ at various temperatures of $(\text{Fe}_{1-x}\text{Co}_x)_4\text{GeTe}_2$ nanoflakes ($x=0.33$) for $H \parallel c$ in 16L (**a**), 7L nanoflakes (**b**) and $(\text{Fe}_{1-x}\text{Co}_x)_4\text{GeTe}_2$ nanoflakes ($x=0.39$) for $H \parallel c$ in 52L (**c**) and 9L nanoflakes (**d**). The $\sigma_{yx}(H)$ curves at low temperatures shows magnetic hysteresis typical for the FM phase in b-d. **e**, Temperature dependence of the magnetic susceptibility $\chi^c(T)$ for $(\text{Fe}_{1-x}\text{Co}_x)_4\text{GeTe}_2$ crystal ($x=0.33$), estimated from the low-field slope of $\sigma_{yx}(H)$ at each temperature (open circle) or from the difference between $\sigma_{yx}(T)$ curves taken at $H = \pm 1$ kOe for $H \parallel c$ (solid circle). Magnetic transition temperature T_N and T_c are indicated by the arrows, together with the characteristic temperature T^* , determined by a broad hump in $\chi^c(T)$. **f**, Thickness dependent magnetic phase diagram of $(\text{Fe}_{1-x}\text{Co}_x)_4\text{GeTe}_2$ crystal ($x=0.33$) with characteristic temperatures of T_N , T^* , and T_c . **g**, Temperature dependence of $\chi^c(T)$ for $x=0.39$ from $\sigma_{yx}(H)$ (open circle) and from magnetization with $H = 1$ kOe for $H \parallel c$ (solid symbol). Characteristic temperature T_0 is indicated by the arrows at the inflection of $\chi^c(T)$ at low temperatures. **h**, Thickness dependent magnetic phase diagram of $x=0.39$ with characteristic temperatures of T_N , T_0 , and T_c .

(Fig.5 and Supplementary Fig. S12). This spin-flop transition is consistent with the bulk case and is a characteristic feature of the AFM phase with the out-of-plane magnetic anisotropy. These results confirm that the AFM phase is stabilized in the nanoflakes with $x = 0.39$, which is more robust than the case of with $x = 0.33$.

In order to show the stable AFM phase in the nanoflakes for $x = 0.39$, we revised Fig. 5 and added a paragraph for describing new results in the main text. Also a new figure (Supplementary Fig. S13) and two additional paragraphs are added in the Supplementary information. In order to emphasize the high- T_N AFM phase in nanoflakes of vdW antiferromagnets, we included the reported T_N of the vdW AFM nanoflakes the Supplementary Table 1.

Q1-5. *The $R(T)$ curves can have many different explanations. I understand the authors did calculations to show the AFM state. However, it is quite often that the calculations results are different from experimental results. From the magnetic and electric transport results, we cannot rule out the material is still in FM state (a FM state with strong in-plane anisotropy). I think that the authors need to do one more experiment, a neutron diffraction on the bulk crystals. If the neutron diffraction confirms the AFM structure, I will fully support the publication of this paper in Nature Communications.*

A1-5.

Due to the current difficult situation, related to the COVID pandemic, we were not able to get an access to the facilities for neutron diffraction experiments. Instead, we carried out resonant soft X-ray scattering experiments on the bulk crystal with $x = 0.33$, which can also offer a direct probe for the superstructure due to the AFM transition. We observed a clear peak with $q = (0\ 0\ 3/2)$, consistent with the calculated interlayer-AFM phase (Supplementary Fig. S7) as explained in **A1-1**. This provides a direct evidence of the AFM phase in $(\text{Fe}_{1-x}\text{Co}_x)_4\text{GeTe}_2$ crystals. Further study using neutron diffraction is desirable to determine the size or orientation of magnetic moments of the observed AFM phase, which, we think, is beyond the scope of this work.

In the revised manuscript, new results on resonant soft X-ray scattering are presented (Supplementary Fig. S7).

Answers to reviewer #2's questions

We sincerely appreciate the reviewer's helpful comments and suggestions. We did our best to answer to each single question and we hope our answers satisfy the reviewer.

Q2-1. *The text does not always refer to the SI when necessary. For many statements in the I had to search for a while to find the data that corroborate that statement. This made reviewing this paper very challenging. I recommend that the authors read their manuscript carefully and make sure they refer the reader to the relevant figure whenever it is described. On a similar note, figure captions can be improved. For example, nowhere in Fig. S2 is the applied field giving, In the main text I found that it is supposedly 1kOe, but in this sentence there is no reference to Figure S2. This is very confusing and hard to follow.*

A2-1. We really appreciate the reviewer's careful reading and useful suggestions. Following the reviewer's suggestion, we carefully revised the manuscript and clearly stated the location of the relevant information (Figures and Supplementary Notes). In addition, we improved the figure captions so that sufficient information is provided to the readers.

Q2-2. *I think Fig 2 should be modified. Fig. 2a does not really show the reader that the phase diagram in Figure 2g is correct. This is better seen with the data from Fig. S2, which I think might be better placed to the main text. In Fig 2a, it appears as if there is a smoot change in magnetization with doping and one cannot see any clean transition from FM to AFM. It took me a while to realize that the magnetic phase diagram is correct. This is also because of the lack of reference to the SI when starting to describing the trends as discussed above. I think in the current stage, the reader will need a long time to understand the results.*

A2-2. As the reviewer suggested, we rearrange graphs in Fig. 2 and Fig. S2 of the main text and the Supplementary information. In the revised manuscript, all the magnetization and susceptibility data as a function of temperature and magnetic field for $(\text{Fe}_{1-x}\text{Co}_x)_4\text{GeTe}_2$ crystals are presented in Fig. 2, in order to emphasize the evolution of the magnetic phase with Co doping. Additional data with different measurement procedures, zero-field-cooling (ZFC) and field-cooling (FC) are also included in Fig. 2 of the revised manuscript. In Fig. 3, we compare the resulting doping-dependent phase diagram with the theoretical calculations. With this arrangement in the revised manuscript, we believe that the readers can understand the results more easily and clearly.

FIG. 2. Doping dependent magnetic phase diagram of $(\text{Fe}_{1-x}\text{Co}_x)_4\text{GeTe}_2$. **a-f**, Temperature dependent magnetic susceptibility $\chi(T)$ for $(\text{Fe}_{1-x}\text{Co}_x)_4\text{GeTe}_2$ ($0 \leq x \leq 0.39$) single crystals, with magnetic field $H = 100$ Oe for different orientations and cooling process, $H \parallel ab$ (open) zero-field-cooling (black), field-cooling (red) and $H \parallel c$ (solid) zero-field-cooling (green), field-cooling (blue). **g**, Magnetization $M(H)$ as a function of magnetic field for $(\text{Fe}_{1-x}\text{Co}_x)_4\text{GeTe}_2$ ($0 \leq x \leq 0.39$) single crystals, for different field orientations, $H \parallel c$ (solid) and $H \parallel ab$ (open). All the $M(H)$ curves were taken at $T = 10$ K, except those for $x = 0.39$ taken at $T = 100$ K. **h**, Temperature dependent in-plane resistivity $\rho(T)$ showing a metallic behavior. **i**, The saturation fields H_{sat} for $H \parallel c$ (solid) and $H \parallel ab$ (open) as a function of Co doping x .

Q2-3. For the higher doped samples, it might be nice to take M vs T curves at more applied fields to better understand the appearance of the unknown magnetic phase.

A2-3. As the reviewer suggested, we performed the temperature dependent magnetization measurements at different magnetic fields ($H = 100$ Oe and 1kOe) below the spin-flop transition field, as shown in Figs. S2 and S3 of the revised Supplementary information. The low-temperature upturn below $T_0 = 75$ K is found to be robust, and a small bifurcation between $\chi(T)$ curves with ZFC and FC was observed. This confirms the low-temperature magnetic structure is distinct from the high-temperature interlayer AFM phase and indicates that the competing FM interaction may play a role. In order to identify the low-temperature magnetic structure of the crystal with $x = 0.39$, the detailed study is needed, which however is beyond the scope of this work.

Following the reviewer #2's suggestion, we added new results of magnetic susceptibility data taken different applied magnetic fields 100 Oe and 1kOe in Supplementary Figs. S2 and S3.

Q2-4. The authors motivate their study with 2D magnets and say there is no 2D metallic AFM with a high Neel temperature yet. That is true. However, the authors also did not introduce a 2D one, since their result suggest that the samples become FM when thin. I think they should be more honest about that fact in the abstract and intro, by inserting the word "bulk" before they discuss their results of a metallic vdW AFM.

FIG. 5. **Thickness-dependent magnetic phase diagram.** **a-d**, Magnetic field dependent transverse conductivity $\sigma_{yx}(H)$ at various temperatures of $(\text{Fe}_{1-x}\text{Co}_x)_4\text{GeTe}_2$ nanoflakes ($x=0.33$) for $H \parallel c$ in 16L (**a**), 7L nanoflakes (**b**) and $(\text{Fe}_{1-x}\text{Co}_x)_4\text{GeTe}_2$ nanoflakes ($x=0.39$) for $H \parallel c$ in 52L (**c**) and 9L nanoflakes (**d**). The $\sigma_{yx}(H)$ curves at low temperatures shows magnetic hysteresis typical for the FM phase in b-d. **e**, Temperature dependence of the magnetic susceptibility $\chi^c(T)$ for $(\text{Fe}_{1-x}\text{Co}_x)_4\text{GeTe}_2$ crystal ($x=0.33$), estimated from the low-field slope of $\sigma_{yx}(H)$ at each temperature (open circle) or from the difference between $\sigma_{yx}(T)$ curves taken at $H = \pm 1$ kOe for $H \parallel c$ (solid circle). Magnetic transition temperature T_N and T_C are indicated by the arrows, together with the characteristic temperature T^* , determined by a broad hump in $\chi^c(T)$. **f**, Thickness dependent magnetic phase diagram of $(\text{Fe}_{1-x}\text{Co}_x)_4\text{GeTe}_2$ crystal ($x=0.33$) with characteristic temperatures of T_N , T^* , and T_C . **g**, Temperature dependence of $\chi^c(T)$ for $x=0.39$ from $\sigma_{yx}(H)$ (open circle) and from magnetization with $H = 1$ kOe for $H \parallel c$ (solid symbol). Characteristic temperature T_0 is indicated by the arrows at the inflection of $\chi^c(T)$ at low temperatures. **h**, Thickness dependent magnetic phase diagram of $x=0.39$ with characteristic temperatures of T_N , T_0 , and T_C .

A2-4. As we discussed in the previous manuscript, the AFM phase of $(\text{Fe}_{1-x}\text{Co}_x)_4\text{GeTe}_2$ nanoflakes with $x = 0.33$ is rapidly suppressed with reducing thickness and is eventually changed to be ferromagnetic for thickness of 7L. This is because the system with $x = 0.33$ is located near to the boundary between FM and AFM phases in the doping-dependent phase diagram, as shown in Fig. 3a of the revised manuscript. Thus we expect that the system with higher Co doping, located deep inside the AFM phase of the phase diagram, would be more stable than the case of $x = 0.33$. In order to confirm this idea, we performed additional measurement on the crystal with $x = 0.39$ and present its thickness-dependence in Fig. 5 of the revised manuscript. As expected, we found that the AFM phase is stable down to ~ 9 L sample with a slightly reduced T_N by $\sim 7\%$ than the bulk value. These additional results indicate that the AFM phase can be stabilized in $(\text{Fe}_{1-x}\text{Co}_x)_4\text{GeTe}_2$ nanoflakes with higher Co doping.

Following the reviewer suggestion, we clearly distinguish the properties of bulk and nanoflakes in abstracts and introduction in order to be more precise on our claim in the revised manuscript.

Q2-5. Can the authors comment on the upturn in resistivity at lower T ? Why is this? Why is it opposite to the trend in FM samples? I think the authors should discuss the role of disorder in their samples and how this might affect the properties. This might also help for the discussion in respect to point 5.

A2-5. We really appreciate the reviewer for the valuable comments. As the reviewer pointed out, all the crystals exhibit a metallic behavior at high temperatures, whereas they show the upturn of the resistivity at low temperatures. We found that the resistivity at low temperature exhibits the $-\ln T$ dependence, followed by deviation at lower temperatures for $(\text{Fe}_{1-x}\text{Co}_x)_4\text{GeTe}_2$ crystals with $x = 0.17, 0.23, 0.26,$ and 0.33 , as presented in the supplementary Fig. S8. This behavior is a characteristic feature of the Kondo scattering, which has been observed in various AFM thin films with substitutional impurities with magnetic moments [for example, D. Khadka *et al. Sci. Adv.* **6**, eabc1977 (2020)]. Recently a metastable phase of $\text{Fe}_{5-x}\text{GeTe}_2$ has been reported, which contains excess of Fe atoms at the interstitial sites above or below the Ge atoms in the unit cell [May, A. F. *et al. ACS Nano*, **13**, 4436-4442 (2019), Stahl, J. *et al. Z. Anorg. Allg. Chem.* **644**, 1923-1929 (2018)]. Considering a smaller size of Co than Fe, a small amount of Co atoms can occupy these interstitial sites in our crystals, which behave as magnetic impurities. These magnetic impurities are coupled the conduction band through exchange interaction, resulting in the Kondo scattering. The excess of the resistivity $\Delta\rho(T) = \rho(T) - \rho_{\min}(T_K)$ for $(\text{Fe}_{1-x}\text{Co}_x)_4\text{GeTe}_2$ crystals are nicely scaled as a function of the normalized temperature by the characteristic temperature T_K with the resistivity minimum, which is consistent with the prediction of Kondo scattering model as shown in Supplementary Fig. S8.

As the reviewer #2 suggested, we added Fig. S8 and a paragraph in the Supplementary information in order to present detailed analysis on the resistivity upturn at low temperatures,

FIG. S8. **Kondo scattering in $(\text{Fe}_{1-x}\text{Co}_x)_4\text{GeTe}_2$.** **A**, Temperature dependent resistivity $\rho(T)$ for $x = 0.17, 0.23, 0.26,$ and 0.33 . Clear low-temperature upturn is commonly observed, which is described by $-\ln T$ (dashed lines) in the intermediate temperature range. **b**, Excess of the resistivity $\Delta\rho(T) = \rho(T) - \rho_{\min}(T_K)$ as a function of normalized temperature T/T_K , due to Kondo scattering. T_K is the characteristic temperature with the resistivity minimum ρ_{\min} . The predicted $\Delta\rho(T/T_K)$ curve (grey solid line) of Kondo scattering model is presented for comparison.

Q2-6. *For the calculations, the authors say they used VCA to model the doping, but they do not say how the lattice constants were obtained. Did they use experimental ones? And if yes in accordance with change in doping? And if not did they optimize the structures? And if yes, did they correct for vdW forces? A few more details here would be helpful.*

A2-6. For the calculation, we used experimental lattice constants as mentioned in Methods section briefly, and they are obtained from the X-ray diffraction results shown in Supplementary Fig. S1c. It can be found that lattice constants change linearly with doping, and thus in the case of higher Co doping than experiments, we set the lattice constants by linear extrapolation. Given lattice constants, we did internal atomic position relaxation for each doping. During internal relaxation, vdW correction is not considered, because internal relaxation is hardly affected by vdW correction term for the thick slab of Fe₄GeTe₂ structure.

In the Methods section of the revised manuscript, we included additional information for calculations, as mentioned above.

Answers to reviewer #3's questions

We sincerely appreciate the reviewer's helpful comments and suggestions. We did our best to answer to each single question and we hope our answers satisfy the reviewer.

Q3-1. Authors have synthesized single crystals, apparently. Therefore, I am puzzled why anisotropy in magnetic properties has not been presented?

A3-1. We presented the magnetization curves at different magnetic fields $H \parallel c$ and $H \parallel ab$ in Fig. 2. As emphasized in the manuscript, depending on Co doping, $(\text{Fe}_{1-x}\text{Co}_x)_4\text{GeTe}_2$ exhibits four different magnetic phases, FM with the out-of-plane magnetic anisotropy ($x = 0$), FM with in-plane anisotropy ($0.07 \leq x < 0.26$), AFM with in-plane magnetic anisotropy ($x = 0.3$ or 0.33), AFM with perpendicular magnetic anisotropy ($x = 0.39$). Particularly the magnetic anisotropy of the AFM phases, for example, the in-plane ($x = 0.33$) and the out-of-plane ($x = 0.39$) anisotropy remain the same in nanoflakes as shown in Fig. 5 of the revised manuscript. These results demonstrate that $(\text{Fe}_{1-x}\text{Co}_x)_4\text{GeTe}_2$ can provide vdW magnetic layers with various spin configuration and orientations, which may be suitable for vdW-material-based spintronic devices.

FIG. 2. Doping dependent magnetic phase diagram of $(\text{Fe}_{1-x}\text{Co}_x)_4\text{GeTe}_2$. **a-f**, Temperature dependent magnetic susceptibility $\chi(T)$ for $(\text{Fe}_{1-x}\text{Co}_x)_4\text{GeTe}_2$ ($0 \leq x \leq 0.39$) single crystals, with magnetic field $H = 100$ Oe for different orientations and cooling process, $H \parallel ab$ (open) zero-field-cooling (black), field-cooling (red) and $H \parallel c$ (solid) zero-field-cooling (green), field-cooling (blue). **g**, Magnetization $M(H)$ as a function of magnetic field for $(\text{Fe}_{1-x}\text{Co}_x)_4\text{GeTe}_2$ ($0 \leq x \leq 0.39$) single crystals, for different field orientations, $H \parallel c$ (solid) and $H \parallel ab$ (open). All the $M(H)$ curves were taken at $T = 10$ K, except those for $x = 0.39$ taken at $T = 100$ K. **h**, Temperature dependent in-plane resistivity $\rho(T)$ showing a metallic behavior. **i**, The saturation fields H_{sat} for $H \parallel c$ (solid) and $H \parallel ab$ (open) as a function of Co doping x .

Q3-2. Supplement section Fig. S1a: crystals have small impurity phase, between the strongest peak around $2\theta = 27$ and (221) there is another reflection in all crystals. Then, $x = 0.39, 0.3, 0.17$ also contain another visible peak between (221) and (444) . Since many results in this work

FIG. S1. **Material synthesis.** **a**, Powder X-ray diffraction patterns of $(\text{Fe}_{1-x}\text{Co}_x)_4\text{GeTe}_2$. Bragg peaks for the phase of Fe_4GeTe_2 are indicated by black triangles. **b**, Bragg peaks of (-101) for polycrystalline $(\text{Fe}_{1-x}\text{Co}_x)_4\text{GeTe}_2$. **c**, In-plane (black) and out-of-plane (red) lattice parameters obtained from Bragg peaks of Fe_4GeTe_2 phase in powder X-ray diffraction patterns of polycrystalline $(\text{Fe}_{1-x}\text{Co}_x)_4\text{GeTe}_2$. **d**, X-ray diffraction patterns for $(\text{Fe}_{1-x}\text{Co}_x)_4\text{GeTe}_2$ ($0.0 \leq x \leq 0.39$) single crystals.

rest on bulk magnetic characterization it is of interest to index those peaks and, if due to impurity phases, ensure that bulk magnetic properties are not influenced by impurities.

A3-2. We appreciate the reviewer for careful reading on our manuscript. All the peaks, that the reviewer 3 mentioned, are well indexed by the single phase of $(\text{Fe}_{1-x}\text{Co}_x)_4\text{GeTe}_2$. We added the corresponding Bragg indices in Supplementary Fig. S1a of the revised manuscript.

Q3-3. *Authors should show saturated moment in $\mu_B/F.U.$ where $F.U.$ is formula unit in Fig. 2a in the main text as well as in Fig. S3(a-e). There seems to be a misprint in Fig. 2a, 0.7 is in fact 0.07.*

A3-3.

We appreciate the careful reading of our manuscript. Following the reviewer suggestion, we corrected the typos and also present the saturation magnetization in terms of $\mu_B/f.u.$ in Fig. 2 and Fig. S6 of the revised manuscript.

Q3-4. *I would also show ZFC-FC $\chi(T)$ curves. Can authors rule out spin glass state in all investigated crystals?*

A3-4. As the reviewer #3 pointed out, the transition to the spin cluster or spin-glass phase can produce a magnetic susceptibility peak, which can be misinterpreted as a signature of the AFM phase transition. In order to rule out the possibility of the spin-cluster state, we measured the temperature dependent magnetic susceptibility, $\chi(T)$, at different cooling procedures, zero-field cooling (ZFC) and field cooling (FC). It has been well known that for the spin-cluster state, the peak of $\chi(T)$ is accompanied by a pronounced bifurcation between ZFC and FC curves. In the $(\text{Fe}_{1-x}\text{Co}_x)_4\text{GeTe}_2$ single crystal with $x = 0.33$, however, we observed negligible difference between the ZFC- and FC- $\chi(T)$ curves for both $H \parallel c$ and $H \parallel ab$ as shown in Fig. 2 of the revised manuscript. These results indicate that the peak in $\chi(T)$ for the crystal with $x = 0.33$ reflects the transition, not to the spin-cluster phase, but to the antiferromagnetic (AFM) phase.

FIG. 2. Doping dependent magnetic phase diagram of $(\text{Fe}_{1-x}\text{Co}_x)_4\text{GeTe}_2$. **a-f**, Temperature dependent magnetic susceptibility $\chi(T)$ for $(\text{Fe}_{1-x}\text{Co}_x)_4\text{GeTe}_2$ ($0 \leq x \leq 0.39$) single crystals, with magnetic field $H = 100$ Oe for different orientations and cooling process, $H \parallel ab$ (open) zero-field-cooling (black), field-cooling (red) and $H \parallel c$ (solid) zero-field-cooling (green), field-cooling (blue). **g**, Magnetization $M(H)$ as a function of magnetic field for $(\text{Fe}_{1-x}\text{Co}_x)_4\text{GeTe}_2$ ($0 \leq x \leq 0.39$) single crystals, for different field orientations, $H \parallel c$ (solid) and $H \parallel ab$ (open). All the $M(H)$ curves were taken at $T = 10$ K, except those for $x = 0.39$ taken at $T = 100$ K. **h**, Temperature dependent in-plane resistivity $\rho(T)$ showing a metallic behavior. **i**, The saturation fields H_{sat} for $H \parallel c$ (solid) and $H \parallel ab$ (open) as a function of Co doping x .

Consistently we also observed no signature of the relaxation behavior, one of the hallmark of the spin cluster state. In the spin cluster phase the change of magnetization requires a sizable time, leading to the time-dependent magnetization phenomena. For example, in the ferromagnetic phase in $(\text{Fe}_{1-x}\text{Co}_x)_4\text{GeTe}_2$ crystals with low x , we found a relaxation behavior with a characteristic time scale of $\tau \sim 600$ sec (Supplementary Fig. S4), which is comparable with the spin cluster phases in, e.g. PrRhSn_3 and $\text{Sr}_2\text{Mn}_{0.7}\text{Fe}_{0.3}\text{MoO}_6$ [Anand, V. K. *et al. Phys. Rev. B* **85**, 014418 (2012), Wang, X. *et al. J. Appl. Phys.* **109**, 07C322 (2011)]. For $x = 0.33$, however, we found no signature of relaxation behavior as shown in Supplementary Fig. S4 in the revised manuscript.

FIG. S4. **Thermoremanent magnetization relaxation in $(\text{Fe}_{1-x}\text{Co}_x)_4\text{GeTe}_2$.** **a**, Magnetic field dependent magnetization for $x = 0.07$ at $T = 10$ K. Clear hysteresis behavior with coercive field $H_c \sim 6$ Oe is observed. **b,c**, Thermoremanent magnetization relaxation for $x = 0.07$ at $T = 10$ K with waiting time $t_w = 193$ s (**b**) and 1350 s (**c**). The red solid line is the best fit of the relaxation model. **d**, Magnetic field dependent magnetization for $x = 0.33$ at $T = 10$ K. In contrast to $x = 0.07$ case, no hysteresis behavior is observed. **e,f**, Thermoremanent magnetization relaxation for $x = 0.33$ at $T = 10$ K with waiting time $t_w = 176$ s (**e**) and 1010 s (**f**). No signature of relaxation behavior is observed for $x = 0.33$.

Finally, in order to present a more direct experimental evidence of the AFM ordering, we performed resonant soft X-ray scattering experiments for the crystal with $x = 0.33$, as shown in the supplementary Fig S7. Similar to neutron diffraction, resonant X-ray scattering also offer a probe for AFM phase. In addition to the Bragg peaks at (003) , a clear additional peak develops below T_N at $q = (0,0,3/2)$, which is absent above T_N . The corresponding periodicity is ~ 19.5 Å along the c -axis, in excellent agreement with the interlayer AFM (A-type) structure, predicted by our first principle calculations. Although further studies, using neutron diffraction, are

FIG. S7. **Resonant soft X-ray scattering of $(\text{Fe}_{0.67}\text{Co}_{0.33})_4\text{GeTe}_2$.** **a**, Resonant soft X-ray scattering intensity at 300 K ($> T_N$) and 70 K ($< T_N$). In addition to Bragg peak at $q = (0\ 0\ 3)$, additional peak at $q = (0,0,3/2)$ is developed below T_N . **b**, Crystal structure of $(\text{Fe}_{0.67}\text{Co}_{0.33})_4\text{GeTe}_2$ with the interlayer AFM structure, indicated by red and blue shades. The period of 19.5 Å, corresponding to $q = (0,0,3/2)$, is consistent with the interlayer AFM phase.

helpful to determine the size and orientation of magnetic moments in the AFM phase of $(\text{Fe}_{1-x}\text{Co}_x)\text{GeTe}_2$, our resonant X-ray scattering results provide a direct evidence of the AFM phase in $(\text{Fe}_{1-x}\text{Co}_x)_4\text{GeTe}_2$ with $x = 0.33$.

We added several sentences and additional Figures (Fig. 2) in the main text and Supplementary Information (Figs. S4, and S7), in order to provide experimental evidence of the long-range AFM phase.

Q3-5. *I find Fig. 1b rather interesting...I wonder what is the Ioffe-Regel limit for AFM crystals? Note the absence of metallic resistivity for high x at low temperatures. Throughout the text authors refer to investigated crystals as “metals”. Yet, $\rho(T)$ looks like a very bad metal for high x. I would advise the use of more precise language.*

A3-5.

We agree with the reviewer about the definition of “metallicity”. A rigorous distinction of metals from insulators is based on the finite resistivity extrapolated at zero temperature. Thus, in principle, metallicity can be defined, even though the resistivity increases with lowering temperatures, which is one of the signatures of ‘bad metal’. As the reviewer #3 suggested, we estimated the mean free path of our crystal with $x = 0.33$. Using the carrier density $n = \sim 6 \times 10^{27} \text{ m}^{-3}$, extracted from the contribution of the normal Hall effect and the measured resistivity we obtained the mean free path of $\sim 2 \text{ nm}$, which is rather close to the in-plane lattice constant $a \sim 0.4 \text{ nm}$. Also from the quantum conductance of the 2D layers and the interlayer distance of $\sim 1 \text{ nm}$ in $(\text{Fe}_{1-x}\text{Co}_x)_4\text{GeTe}_2$ crystals, we obtained a critical resistivity $\rho \sim 200 \mu\Omega\text{cm}$, which is comparable with the measured resistivity. These results show that $(\text{Fe}_{1-x}\text{Co}_x)_4\text{GeTe}_2$ crystals are in the bad metal regime.

In the revised manuscript, we clearly mentioned that our system is in the bad metal regime.

Q3-6. *In Fig S5 authors report change in DOS with Co doping. Therefore, it is reasonable to assume some change in electronic structure. Does that have any effect on electrical detection of the spin state. For example, could changes in AMR be influenced by electronic structure changes, in addition to Neel vector changes?*

A3-6. As the reviewer #3 pointed out, there is electronic structure change due to Co doping, which is expected to affect the magnetotransport properties such as the anisotropic magnetoresistance (AMR). Although the AMR itself is defined by the relative orientation between Neel vector and the current direction, its magnitude is material specific and thus can be significantly modulated by Co doping. For the crystal with $x = 0.33$ the maximum AMR is found to be $\sim 0.3 \%$, comparable with those of other AFM metals. We believe that the size of the AMR can be further optimized by control of Co doping level. Considering the narrow doping window for the in-plane AFM phase in the phase diagram shown in Fig. 3a, however, the detailed study requires precise control of Co doping level, which remains as a future work.

Q3-7. I do not think that this work brings high impact or innovation. I agree that AFM transition is high but the last sentence in the introduction is greatly exaggerated. Promising material in spintronics can not operate at 230 K. Authors also say in the conclusion that (Fe,Co)₄GeTe₂ will enrich the material candidates and spin functionalities for...spintronics. I do not see how. Thickness-dependent magnetic transition falls down with size reduction therefore it is difficult to think of future devices based on nanofabricated crystals investigated in this work.

A3-7. Realization of stable room temperature antiferromagnetism in nanoflakes is one of the challenge in the current research field of vdW magnets. We note that most of vdW antiferromagnets so far investigated, show various spintronic functionalities below T_N much lower than our system. For example, CrI₃, one of the most widely studied vdW magnets, has a $T_N \sim 30$ K, far below room temperature. This contrasts to FM vdW materials, some of which exhibit ferromagnetism nearly at room temperature. Although our (Fe_{1-x}Co_x)₄GeTe₂ shows $T_N \sim 226$ K, below room temperature, this is significant increase as compared to the previously studied vdW antiferromagnets. Unlike the previously studied vdW antiferromagnets with intralayer AFM structures, our approach for high- T_N vdW antiferromagnetism is to use high- T_c FM vdW metals as a parent material and convert them to antiferromagnet by changing their interlayer interaction. We believe that our approach can be applied to other high- T_c FM vdW materials, which will eventually result in higher- T_N antiferromagnetism in the vdW magnets.

We agree with the reviewer that the stability issue of the AFM phase in nanoflakes is even more important than the issue of high- T_N in bulk materials. As the reviewer pointed out, the AFM phase of our crystal with $x = 0.33$ is rapidly suppressed with lowering thickness and eventually changes to be ferromagnetic for thickness of 7 layers. However as explained in the manuscript, this is because the system with $x = 0.33$ is located close to the boundary between FM and AFM phases in the doping-dependent phase diagram, as shown in Fig. 3a of the revised manuscript. Thus we expect that the system with higher Co doping, located deep inside the AFM phase of the phase diagram, would be more stable than the case of $x = 0.33$. In order to confirm this idea, we performed new measurements on the crystal with $x = 0.39$ and presented its thickness-dependence in the revised manuscript. As expected, we found that the AFM phase is stable down to ~ 9 L sample with a slightly reduced T_N , by $\sim 7\%$ than the bulk value. To best our knowledge, this is the highest T_N in the AFM vdW nanoflakes (Supplementary Table. S1).

We envision that our finding of the stable AFM phase in ultrathin (Fe_{1-x}Co_x)₄GeTe₂ nanoflakes leads to more researches on spintronic properties using metallic vdW AFM layers. As demonstrated in one of the FM vdW metal Fe₃GeTe₂ [Deng, Y. *et al. Nature*, **563**, 94-99 (2018)], electrical tuning of the Neel temperature can be possible using (Fe_{1-x}Co_x)₄GeTe₂ nanoflakes. Also heterostructures, consisting of AFM layers of (Fe,Co)₄GeTe₂ and FM layers of Fe₃GeTe₂ or Fe₄GeTe₂, may induce the exchange bias effect, which has been demonstrated by 25-75 layer-thick CrCl₃ and Fe₃GeTe₂ [Zhu, R. *et al. Nano Lett.* **20**, 5030-5035 (2020)]. Thus, we believe that our work will bring attention to AFM vdW nanoflakes as one of the active component in vdW-material-based spintronics.

In order to show the stable AFM phase in the nanoflakes for $x = 0.39$, we revised Fig. 5 and added a paragraph for describing new results in the main text. Also a new figure (Supplementary Fig. S13) and two additional paragraphs are added in the Supplementary information. In order to emphasize the high- T_N AFM phase in nanoflakes of vdW antiferromagnets, we included the reported T_N of the vdW AFM nanoflakes the Supplementary Table 1.

List of the changes

1. We included four authors, Woo-suk Noh, J. Y. Kim, Gyuseung Choi, and J. -H. Park, who performed resonant soft X-ray scattering and additional transport property experiments. Also, the content in Author Contributions is changed accordingly.

2. In **Methods**, we added one paragraphs describing the experimental details about resonant X-ray scattering. Also we included additional information for first principle calculations.

[Methods, Line 314-318]

“Experimental lattice constants of $(\text{Fe,Co})_4\text{GeTe}_2$, obtained from the X-ray diffraction...We did internal atomic position relaxation for each doping without vdW correction.”

[Methods, Line 322-326]

“Resonant soft X-ray scattering. Resonant soft X-ray scattering experiment was carried out... have different energy profile with (0 0 3) Bragg peak.”

3. As the reviewer #1 and #3 requested, we added several sentences and additional Figures in the main text and Supplementary Information, in order to provide experimental evidence of the long-range AFM phase. [A1-1, A1-5, A3-4]

[Results, Line 117-118]

“Negligible bifurcation between $\chi(T)$ curves taken during ZFC and FC is consistent with the long-range AFM phase.”

[Results, Line 139-141]

“Furthermore, from magnetic X-ray scattering experiments, we observed the magnetic Bragg peak at $q = (0\ 0\ 3/2)$, developed below T_N for the crystal with $x = 0.33$. This confirms the A-type AFM structure, as predicted by calculations (Supplementary Fig. S7).”

[Figure 2]

Additional graphs (a-f) of temperature dependent magnetic susceptibility data for the crystals with different doping are presented.

[Supplementary Fig. S4]

New results on thermoremanent magnetization relaxation are presented.

[Supplementary Fig. S7]

New results on resonant soft X-ray scattering are presented.

4. In order to show the stable AFM phase in the nanoflakes for $x = 0.39$, we revised Fig. 5 and added a paragraph for describing new results in the main text. Also a new figure and two additional paragraphs are added in the Supplementary information. [A1-4, A2-4]

[Figure 5]

In addition to the results for $x = 0.33$, new results on $x = 0.39$ are added.

[Results, Line 227-240]

“In nanoflakes with higher Co doping of $x=0.39$, located deep inside... controlled by thickness and doping levels in $(\text{Fe}_{1-x}\text{Co}_x)_4\text{GeTe}_2$.”

[Supplementary Fig. S13]

New results on nanoflakes with $x=0.39$ are presented.

[Last two paragraphs in the Supplementary information]

“The reason why the magnetic phase of the crystal... This stable AFM phase in nanoflakes with $x=0.39$ is in strong contrast to the case of $x=0.33$.”

“While the transition temperatures T_N and T_0 are not sensitive to... the detailed magnetic properties can be tune by controlling thickness.”

5. As the reviewers #2 and #3 suggested, we presented the temperature-dependent susceptibility results, together with other magnetic properties in Fig. 2 of the revised main text. Accordingly, the phase diagram and the calculation results are shown in Fig. 3. [A2-2, A3-4]

6. Following the reviewer #2's suggestion, we added new results of magnetic susceptibility data taken different applied magnetic fields 100 Oe and 1kOe in Supplementary Figs. S2 and S3. [A2-3]

7. Following the reviewer #2's suggestion, we clearly distinguish the properties of bulk and nanoflakes in abstracts and introduction in order to be more precise on our claim in the revised manuscript. [A2-4]

[Abstract, Line 35-36]

“This leads to high- T_N antiferromagnetism of $T_N \sim 226$ K in a bulk and ~ 210 K in nine-layer-thick nanoflakes...”

[Introduction, Line 67-68]

“... hosts the interlayer AFM phase with $T_N \sim 226$ K in a bulk and ~ 210 K in nine-layer-thick nanoflakes.”

8. Additional Figure S8 and a paragraph are added in the Supplementary information in order to present detailed analysis on the resistivity upturn at low temperatures, as the reviewer #2 suggested. [A2-5]

[Supplementary Fig. S8]

New results on the resistivity upturn and the Kondo scattering model are presented.

[Second paragraph in Supplementary Note 2.]

“Furthermore, at low temperatures, we found that the resistivity exhibits ...which is consistent with the prediction of Kondo scattering model as shown in Supplementary Fig. S8.”

9. As requested by the reviewer #3, we revised the Supplementary Fig. S1 to show that all the peaks are described by the single phase of $(\text{Fe,Co})_4\text{GeTe}_2$. [A3-2]

10. Following the reviewer #3's suggestion, we change the unit for the magnetization in Fig. 2 and Fig. S6. [A3-3]

11. Bad metal behavior is clearly mentioned in the main text and the Supplementary information as the reviewer #3 suggested. [A3-5]

[Results, Line 91-92]

“... which is comparable with that of the pristine Fe_4GeTe_2 and in the bad metal regime (Supplementary Fig. S8).”

[First paragraph in Supplementary Note 2.]

“As shown in Fig. 2 of the main text, all the crystals exhibit a metallic behavior... These results show that $(\text{Fe}_{1-x}\text{Co}_x)_4\text{GeTe}_2$ crystals are in the bad metal regime.”

12. In order to emphasize the high- T_N AFM phase in nanoflakes of vdW antiferromagnets, we included the reported T_N of the vdW AFM nanoflakes the Supplementary Table 1. [A3-7]

Reviewers' Comments:

Reviewer #1:

Remarks to the Author:

In the revised manuscript, the authors provide more experimental results, such as ZFC/FC, relaxation measurements and soft X-ray. I now believe that the material in bulk state can become antiferromagnetic with certain Co doping. The material system is quite interesting because very few van der Waals AFM has such a high Neel temperature. I also found that the $x = 0.39$ sample is extremely interesting. Thin flake of $x = 0.39$ can change from FM to AFM with increasing temperature. Interesting vdW heterostructures can be fabricated using $x = 0.39$. $x = 0.33$ is not that interesting, because it becomes FM in the form of thin flake. I support to publish this paper in Nature Comm.

Reviewer #3:

Remarks to the Author:

Authors have presented a very detailed response to all referee questions. I find this rather persuasive and would like to congratulate authors on work well done.

Authors state quite properly that magnetic order transition temperature is substantially higher when compared to other materials in this class, yet it is still below the room temperature.

I recommend publishing in the present form.

Reply to the Reviewers

Answers to reviewer #1's questions

We sincerely appreciate the reviewer's helpful comments and suggestions. We did our best to answer to each single question and we hope our answers satisfy the reviewer.

Q1-1. *In the revised manuscript, the authors provide more experimental results, such as ZFC/FC, relaxation measurements and soft X-ray. I now believe that the material in bulk state can become antiferromagnetic with certain Co doping. The material system is quite interesting because very few van der Waals AFM has such a high Neel temperature. I also found that the $x = 0.39$ sample is extremely interesting. Thin flake of $x = 0.39$ can change from FM to AFM with increasing temperature. Interesting vdW heterostructures can be fabricated using $x = 0.39$. $x = 0.33$ is not that interesting, because it becomes FM in the form of thin flake. I support to publish this paper in Nature Comm.*

A1-1. Thank you very much for reviewer #1's support to publish our paper in *Nature Communications*. We agree with reviewer that the sample with $x = 0.39$ is extremely interesting. We will further investigate on magnetic properties of bulk crystals and thin flakes which will be reported as a separate work.

Answers to reviewer #3's questions

We sincerely appreciate the reviewer's helpful comments and suggestions. We did our best to answer to each single question and we hope our answers satisfy the reviewer.

Q3-1. *Authors have presented a very detailed response to all referee questions. I find this rather persuasive and would like to congratulate authors on work well done. Authors state quite properly that magnetic order transition temperature is substantially higher when compared to other materials in this class, yet it is still below the room temperature. I recommend publishing in the present form.*

A3-1. Thank you very much for reviewer #3's support to publish our paper in *Nature Communications*. We hope our work enrich the material candidates and the spin functionalities for all-vdW-material-based spintronics.